# SingRef6D: Monocular Novel Object Pose Estimation with a Single RGB Reference

**Jiahui Wang**[1]    **Haiyue Zhu**[2][†]    **Haoren Guo**[1]    **Abdullah Al Mamun**[1]
**Cheng Xiang**[1]    **Tong Heng Lee**[1]

[1]College of Design and Engineering, National University of Singapore
[2]SIMTech, Agency for Science, Technology and Research (A*STAR)
wjiahui@u.nus.edu   zhu_haiyue@simtech.a-star.edu.sg

## Abstract

Recent 6D pose estimation methods demonstrate notable performance but still face some practical limitations. For instance, many of them rely heavily on sensor depth, which may fail with challenging surface conditions, such as transparent or highly reflective materials. In the meantime, RGB-based solutions provide less robust matching performance in low-light and texture-less scenes due to the lack of geometry information. Motivated by these, we propose **SingRef6D**, a lightweight pipeline requiring only a **single RGB** image as a reference, eliminating the need for costly depth sensors, multi-view image acquisition, or training view synthesis models and neural fields. This enables SingRef6D to remain robust and capable even under resource-limited settings where depth or dense templates are unavailable. Our framework incorporates two key innovations. First, we propose a token-scaler-based fine-tuning mechanism with a novel optimization loss on top of Depth-Anything v2 to enhance its ability to predict accurate depth, even for challenging surfaces. Our results show a 14.41% improvement (in $\delta_{1.05}$) on REAL275 depth prediction compared to Depth-Anything v2 (with fine-tuned head). Second, benefiting from depth availability, we introduce a depth-aware matching process that effectively integrates spatial relationships within LoFTR, enabling our system to handle matching for challenging materials and lighting conditions. Evaluations of pose estimation on the REAL275, ClearPose, and Toyota-Light datasets show that our approach surpasses state-of-the-art methods, achieving a 6.1% improvement in average recall. Project page: https://plusgrey.github.io/singref6d.

## 1 Introduction

Determining the 6D pose of an object in a three-dimensional space is an essential task for various applications in robotics, industrial automation, and augmented reality [1–3]. Recent computer vision approaches for 6D pose estimation have achieved remarkable progress, enabling machines to understand object orientation and position in space and interact with the physical world with increasing sophistication [4–10].

Among these methods, one prevalent paradigm relies on accurate geometric information of object CAD models and scene depth maps. By matching observed scenes with pre-obtained 3D models, it achieves notable 6D pose estimation performance [6, 8, 9, 11, 12]. Although these solutions have shown significant effectiveness, they come with some practical limitations. First, obtaining CAD models for new objects is costly, requires specialized scanning equipment, and involves tedious

---

[†]Corresponding author.

39th Conference on Neural Information Processing Systems (NeurIPS 2025).

manual refinement. In addition, sensor-based depth sensing struggles with challenging surface conditions, particularly for black, transparent, and highly reflective materials. Our study shows that it leads to a failure rate exceeding **85**% in the ClearPose [13] dataset that focuses on scenes with transparent objects. Alternatively, some methods use multiview image matching [7, 4, 5] for 6D pose estimation, either from object projections [10, 4], video sequences [9, 14] or leveraging neural fields [15] for view rendering. However, multiview matching relies on an extensive template library for high accuracy, while constructing neural fields is computationally intensive and restricted to per-instance training.

In contrast, the human visual system excels at object pose estimation with remarkable efficiency and adaptability across diverse environments. It operates without explicit 3D models, exhaustive viewpoint sampling, or even strict binocular vision, instead relying on cognitive mechanisms for depth perception and shape understanding [16–22]. Inspired by the human vision system, we propose *SingRef6D*, a simple-yet-efficient pose estimation pipeline that requires only a **Sing**le **Ref**erence RGB image, eliminating the need for explicit 3D models, precise depth sensing, or any form of novel view synthesis, explicit or implicit, while maintaining robustness and versatility. Moreover, Singref6D requires neither the training of costly generative models (e.g., VAE [23], diffusion [24]) nor the construction of scene-specific neural fields (e.g., NeRF [25]), yet achieves competitive performance.

Specifically, *SingRef6D* tackles the pose estimation problem by independently addressing their corresponding challenges in both depth perception and pose solving stages. To enhance depth perception, we develop a new fine-tuning approach for Depth-Anything v2 (DPAv2) [26] by using a token scaler (a network to re-weight features from transformer layers) to scale hierarchical features dynamically. This refinement mimics the mechanism of human spatial perception in a stratified manner [20, 22], allowing our depth prediction module to reliably extract depth cues from a single RGB image, even under adverse conditions such as high reflectivity and transparency. Although our depth model is trained with supervision, it effectively distills spatial priors into a compact model, enabling inference-time operation with only an RGB input. In this sense, our method implicitly expands the reference space without incurring the cost of dense geometry or view synthesis. With LoFTR's strengths [27], our second stage introduces a depth-aware matching module by fusing RGB and depth cues into a unified latent space. By encoding spatial priors during training, our depth model expands the effective view space, allowing LoFTR [27] to match across challenging appearances with only a single-view reference. This integrated approach significantly refines pose estimates and ensures precise alignment even under challenging environmental conditions. The contributions of this work can be summarized as follows:

- We introduce **SingRef6D**, a novel monocular 6D pose estimation pipeline requiring only a single reference RGB image under a strictly minimal reference setup, without relying on CAD models, multi-view collections, or novel-view synthesis .

- We developed a token-scaler-based fine-tuning approach for DPAv2 [26], which enables our metric depth estimation to handle challenging surface conditions. Results on the ClearPose dataset [13] show a boosted accuracy from 31.23% to 54.30% for transparent objects.

- We propose a depth-aware matching on top of LoFTR [27], and our results show an improvement of 6.1% in average recall across three pose estimation benchmarks.

## 2 Related Work

### 2.1 Novel Object 6D Pose Estimation

Novel object 6D pose estimation aims to identify the pose of a previously unseen object. Current methods typically adhere to the following pipeline: identification of the object, integration of 3D information, matching, and pose solving. These methods fall into two main splits, one is feature matching-based methods [28–30, 8, 11, 31, 12, 32, 4, 33, 9, 10, 6] the other is template matching-based methods [34, 14, 5, 35, 7, 4, 36–39]. For each category, we investigate several representative models and discuss their limitations.

**Feature-Based**. MatchU [32] simultaneously extracts features from RGB, depth, and CAD models. Then, fine-grained matching with the decoded descriptor will be conducted. However, its heavy dependence on CAD models and substantial training overhead limit its practicality in real-world

Table 1: Comparison of input data requirement: (1) Reference Required: the format and utilization of input data as reference required for pose estimation; (2) Extra cost: cost to obtain sufficient spatial information; and (3) Localization: the method for object localization.

| Method | Reference Required | Extra Cost | Localization |
|---|---|---|---|
| OVE6D [39] | Precise CAD models | - | Mask |
| MegaPose [40] | Precise CAD models | - | Bounding Box |
| Gen6D [41] | RGB video sequence | Image-level Comparison | Bounding Box |
| OnePose [9] | RGB video sequence | Structure from Motion (SfM) | Bounding Box |
| NOPE [7] | Single RGB image and poses of new viewpoints | Train a U-Net to synthesis 342 novel views with VAE | (One object scene) |
| PoseDiffusion [42] | Multiview RGB templates | Train a diffusion network to generate poses | Bounding Box |
| SAM6D [4] | Multiview RGBD templates | Train a 3D-based matching model | Mask |
| FoundationPose [5] | Multiview RGB templates | Train neural fields to represent 3D spaces | Mask |
| Oryon [6] | RGBD image and text prompt | Train a matching and a segmentation model | Mask |
| Any-6D [43] | Single RGBD image | Image-to-3D model | Mask |
| Zero123-6D [44] | RGB images | Synthesize 50 novel views with diffusion model, train neural fields to represent 3D spaces | (One object scene) |
| 3DAHV,DVMNet [45, 46] | Single RGB image | **Large-scale training** for a 2D-3D latent space | Mask |
| SingRef6D (ours) | Single RGB image | Fine-tune a **lightweight** token scaler (only for depth) | Mask |

applications. Oryon [6] uses a vision-language model to boost matching with text embedding. This approach fails to deal with transparent objects due to the invalid values of sensor-based depth.

**Template-Based**. NOPE [7] determines the pose of the object by image-level matching that requires extensive viewpoints to synthesize templates from the reference image to represent 3D space adequately. FoundationPose [5] trains neural fields, such as BundleSDF [15], to generate a number of pose hypotheses. However, NOPE [7] is limited to scenarios involving a single textured object, as it predicts the object's appearance from novel viewpoints. This approach becomes incapable for complex scenes with multiple objects and suffers significant performance degradation when the viewpoints are spatially sparse. Such a reliance on view synthesis limits generalization to multi-object or low-texture sceneschallenges that SingRef6D explicitly tackles via a learned depth prior. The hypothesis generation with refinement in FoundationPose [5] incurs computational overhead, and the neural field may fail when facing challenging light conditions.

Table 1 summarizes the data requirements of different models for novel object 6D pose estimation. Although some models [7, 38, 44] **appear** to use a single RGB image as a reference, they incorporate techniques such as diffusion-based synthesis or 3D reconstruction. For instance, NOPE [7] trains a network to synthesize image embeddings of novel views and requires inputting the pose of new viewpoints. Zero123-6D [44] trains a NeRF-like [25] network to reconstruct the 3D shape with new views generated by a diffusion model. GigaPose [38] directly uses an image-to-3D model to obtain a mesh from an image, which might result in large shape inaccuracy.

Our SingRef6D, in contrast, does not involve any format of multi-view rendering or novel view synthesis; all accessible reference information comes from a single RGB image. This is one of the minimal and strictest requirements for the reference input that ensures strong adaptability even in data-scarce and resource-scarce scenarios. This sharply contrasts with approaches that require synthesizing dozens or hundreds of virtual views, which incur high training and inference costs. Such an efficiency makes it well-suited for real-world applications where limited annotations or sparse reference data are available.

### 2.2 Monocular Metric Depth Estimation

We investigate several monocular metric depth estimation methods [47–52, 26, 53]. MiDas [47], the first transformer-based dense depth model, introduced top-to-bottom fusion for precise predictions. ZoeDepth [48] requires extensive fine-tuning for accurate zero-shot depth estimation. UniDepth [49] employs a self-promptable camera module for depth conditioning but struggles with generalization due to full supervision. ScaleDepth [50] models metric-relative depth relationships based on the assumption that all transformations are linear, which has achieved efficiency but relatively low accuracy. HybridDepth [51] performs well across various datasets but relies on an additional focal stack for spatial priors. Depth-Anything v2 [26], structurally similar to MiDas [47], benefits from class-agnostic self-supervised training, ensuring adaptability and fine-tuning ease. Given these insights, we fine-tune a pre-trained DPAv2 [26] as our depth estimator. Details of our pipeline are provided in Section 3.

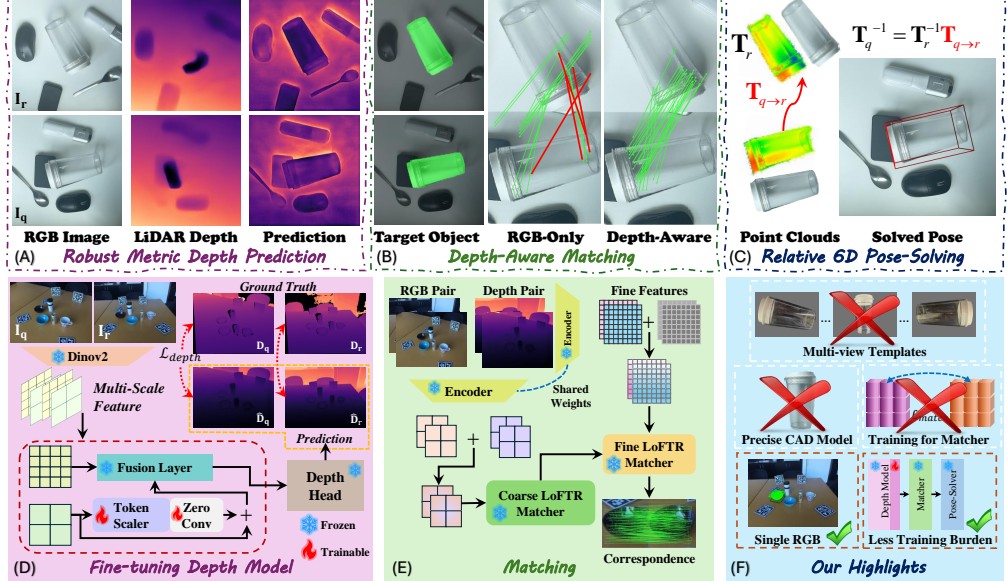

Figure 1: Visualized pipeline for inference (**upper**), the details of our depth model, matching process, and highlights (**lower**). During inference, our fine-tuned depth model first estimates the metric depth accurately, which can deal with challenging surfaces. Subsequently, the proposed depth-aware matching utilizes depth value as spatial cues to establish correspondences even in low-textured regions. Then, the relative pose $\mathbf{T}_{q \rightarrow r}$ can be solved with a point cloud registration model and the 6D pose for the query object can be calculated with $\mathbf{T}_q^{-1} = \mathbf{T}_r^{-1}\mathbf{T}_{q \rightarrow r}$.

## 3  Method

### 3.1  Overview

The 6D pose estimation problem addressed in this work is defined as follows. Given a pair of RGB images, a query, and a reference, both containing the same object, the objective is to estimate the 6D relative pose between the query and reference. Unlike previous methods [4, 10, 9], our approach does not rely on CAD shapes, multi-view images, or any extra costly training to implicitly obtain novel-view information. This **"low"** reference requirement improves the applicability in real-world scenarios, making our method highly practical and versatile.

Figure 1 outlines our pipeline, which begins by processing query and reference image pairs to generate depth predictions using our robust depth prediction module (Figure 1A). Next, query-reference correspondence is established through the depth-aware matching module (Figure 1B). With the availability of off-the-shelf models like SAM [54], target object localization is simplified, enabling correspondence by cropping the scene to a region of interest. After that, we compute the 3D relative pose using PointDSC [55] and depth-projected point clouds.

### 3.2  Robust Metric Depth Prediction

Accurate geometry perception relies on precise metric depth estimation. While depth foundation models, such as DPAv2 [26], provide an effective solution for depth prediction, we observed that direct metric depth estimation from DPAv2 [26] is hindered by scaling inconsistencies. To address this limitation, fine-tuning can be employed to mitigate the scaling issue. However, direct fine-tuning is either computationally demanding or often degrades clarity, leading to distorted boundaries that diminish overall performance, as illustrated in Figure 2. To overcome these challenges, we introduce a novel fine-tuning approach based on a token-scaler mechanism, designed to enhance the accuracy and robustness of metric depth prediction based on DPAv2 [26]. Our approach integrates a ControlNet-like structure [56] to dynamically scale and modulate features at different levels (low-level, mid-level, high-level, and global-level) of DPAv2 [26]. As shown in Figure 3, the depth prediction pipeline incorporates the token-scaler mechanism for hierarchical feature modulation,

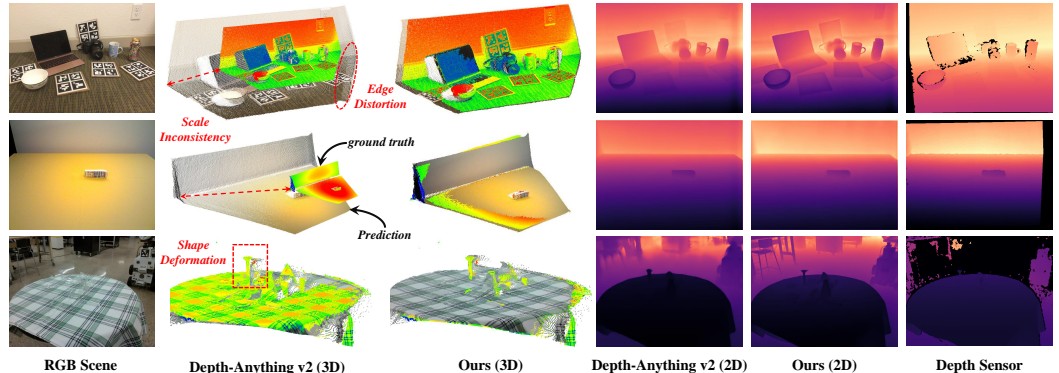

RGB Scene    Depth-Anything v2 (3D)    Ours (3D)    Depth-Anything v2 (2D)    Ours (2D)    Depth Sensor

Figure 2: Visualized comparison of projected point clouds with depth maps. The depth sensors were unable to detect transparent objects and are limited to the device's inherent field of view. The predicted depth ensures a valid value in all pixels. Compared to vanilla DPAv2 [26](with a fine-tuned head), point clouds using depths from our method are geometrically consistent and scale-correct.

which adaptively adjusts feature representations at each level without compromising computational efficiency and pre-training knowledge.

Our method performs well in preserving geometric characteristics and overall depth map quality, as demonstrated in Figure 2. Mathematically, consider a backbone network (e.g., DINOv2 [57] in DPAv2 [26]) for feature extraction, let $\mathbf{F}_l$ denote the features at stage $l$, where $l \in \{1, 2, 3, 4\}$ corresponds to low-level, mid-level, high-level, and global-level features, respectively. We introduce a novel token scaler that adaptively re-weights the feature in each level and then fuses with the upper level as:

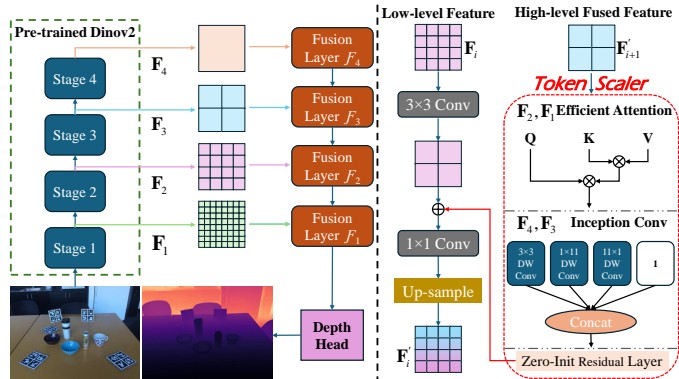

Figure 3: Visualized structure of the depth prediction pipeline (**Left**) and fusion layer with our token scaler (**Right**).

$$\mathbf{F}'_l = \mathcal{F}_l \left( \mathbf{F}_l, Scaler(\mathbf{F}'_{l+1}) \right), \quad l \in \{1, 2, 3\} \tag{1}$$

where $Scaler(\cdot)$ is a scale function, $\mathcal{F}_l$ is the fusion *conv* layer and $\mathbf{F}'_4 = \mathrm{Upsample}(Scaler(\mathbf{F}_4))$. Specifically, for low- and middle-level features ($\mathbf{F}_1$ and $\mathbf{F}_2$), which retain fine-grained details and local features with high-frequency information, we apply an efficient attention layer [58] to boost global awareness while minimizing the impact of potential noise. Conversely, for high- and global-level features ($\mathbf{F}_3$ and $\mathbf{F}_4$), which primarily capture low-frequency global scene and context-level information, we employ an InceptConv-based network [59] as the scaler function, emphasizing local features to enhance the high-level feature map. More details are illustrated in Sections C.1 and C.2 of the Appendix.

In addition to the token-scaler mechanism, our fine-tuning approach also employs a novel loss scheme that combines two key components: a global loss term to regulate the scale and shift of the depth map and a local loss term to improve object geometry and surface reconstruction, i.e., $\mathcal{L}_{depth} = \mathcal{L}_{local} + \mathcal{L}_{global}$.

**Global Loss**. We build on the Scale-Shift Invariant (SSI) Loss with regularization, denoted as $\mathcal{L}_{ssi} + \mathcal{L}_{reg}$. It has proven effective in models like MiDaS [47] and DPAv2 [26] for overall supervision. To further enhance performance, we incorporate a BerHu [60] loss as it complements SSI by better penalizing large residuals, especially in high-error regions (default loss in many depth estimation models), and the final global loss is,

$$\mathcal{L}_{global} = \mathcal{L}_{ssi} + \mathcal{L}_{reg} + \alpha \, \mathrm{BerHu}(\mathbf{D}, \hat{\mathbf{D}}), \tag{2}$$

where $\alpha$ is a hyper-parameter, $\mathbf{D}$ is the ground truth depth and $\hat{\mathbf{D}}$ is the prediction. The detailed mathematical expression of our global loss is illustrated in the supplementary material Section C.3.

The local loss consists of three parts, $\mathcal{L}_{local} = \mathcal{L}_{scale} + \mathcal{L}_{edge} + \mathcal{L}_{norm}$., where each is described in the following.

**Scale Alignment Loss**. The SSI loss [47] effectively handles global scale and shift parameters but does not directly enforce object-level scale alignment, potentially reducing the accuracy of relative scale representations. To address this, we introduce a scale alignment loss

$$\mathcal{L}_{scale} = \frac{1}{M} \sum_i \frac{(\hat{d}_i - d_i)^2}{1 + \eta|\hat{d}_i - d_i|}, \tag{3}$$

where $d_i$ and $\hat{d}_i$ denotes the ground truth and predicted depth value of the $i$-th pixel, respectively. This loss quantifies the discrepancy between the ground truth and the prediction within an object. To enhance robustness against outliers, we introduce an extra term with $\eta$ into the loss. For large errors, it reduces the gradient, mitigating noise sensitivity, while for small errors, it approximates MSE, enabling flexible supervision.

**Edge-emphasize Loss**. Our analysis indicates that the geometry of the shape is predominantly determined by its edges, whereby inaccurate edge reconstruction in depth maps results in substantial 3D distortions, as illustrated in Figure 2. Utilizing predictions from DPAv2 [52], the textures are generally acceptable; however, the boundaries exhibit pronounced discontinuities. To this end, we propose an edge-emphasized loss:

$$\mathcal{L}_{edge} = \frac{1}{M} \sum_i e^{-\sigma \|\nabla I_i\|} \cdot \left\| \nabla \hat{d}_i - \nabla d_i \right\|_2^2, \tag{4}$$

where $\nabla I_i$ is the gradient of the corresponding position in the RGB image, and $\sigma$ is the weight. This loss allows for depth variations in regions with sharp texture changes, which typically indicate boundaries while constraining depth variations in areas with locally similar textures.

**Normal Consistency Loss**. While existing loss functions effectively address object scale and edge detection, they fail to account for surface deformation. Although precise edge detection aids object localization, errors in 3D surface projection can still compromise relative pose accuracy. To address this, we introduce a normal consistency loss that preserves geometric structure,

$$\mathcal{L}_{norm} = \frac{1}{M} \sum_{i \in M} e^{-\lambda \|\nabla d_i\|} \cdot (1 - \frac{\left\langle \vec{n}_{\hat{\mathbf{D}}}^i, \vec{n}_{\mathbf{D}}^i \right\rangle}{\left\| \vec{n}_{\hat{\mathbf{D}}}^i \right\| \cdot \left\| \vec{n}_{\mathbf{D}}^i \right\|}). \tag{5}$$

where $\vec{n}$ represent the norm vector. Optimizing this loss enforces directional consistency of surface normals in the predicted depth map, ensuring alignment with ground truth and maintaining surface coherence for more accurate geometric reconstruction. The calculation details of the norm vector are discussed in the supplementary material Section C.4. In our approach, the enhanced depth prediction not only contributes to the geometry perception but also enhances matching performance for shape understanding, as detailed in Section 3.3

### 3.3 Depth-Aware Matching and Pose-Solving

As Figure 1B illustrates, RGB-only matching relies heavily on texture and local brightness, resulting in two potential limitations: frequent mismatches between similarly textured foreground and background regions and poor performance in low-light areas such as shadows. These issues can degrade the accuracy of pose solving by introducing unsatisfactory matches. To address this challenge, we propose a fine-tuning-free depth-aware matching module that effectively combines metric depth with RGB input to enhance spatial context understanding. We extend LoFTR [27] by incorporating corresponding depth maps as additional inputs, which enables the fusion of features between depth and RGB representations in the latent space (as shown in Figure 1E). To preserve the well-trained feature extraction capability of LoFTR [27], we keep its parameters frozen while leveraging its coarse-to-fine matching strategy on the combined features. Section C.5 of the supplementary material provides a detailed mathematical formulation of this process. Finally, we use PointDSC [55] to refine the matched correspondences and estimate the relative pose $\mathbf{T}_{q \to r}$, thus solving the desired 6D pose through $\mathbf{T}_q^{-1} = \mathbf{T}_r^{-1} \mathbf{T}_{q \to r}$.

Table 2: Quantitative result of metric depth estimation on various benchmark datasets. The baselines are fine-tuned with the corresponding training data for fair comparison. Our main results are highlighted with `colored fonts`, and the best result of each column is shown in **bold fonts**.

| Dataset | Method | $\delta_{1.05}\uparrow$ | $\delta_{1.10}\uparrow$ | $\delta_{1.25}\uparrow$ | RMSE↓ | $log10\downarrow$ | Abs.Rel.↓ | Sq.Rel.↓ | MAE↓ |
|---|---|---|---|---|---|---|---|---|---|
| Toyota-Light [62] | Unidepth (FT) | 11.80 | 31.24 | 60.85 | 0.422 | 1.022 | 1.640 | 0.557 | 0.371 |
| | Depth-Anything v2 (FT) | 14.64 | 33.65 | 64.23 | 0.264 | 0.249 | 0.252 | 0.085 | 0.211 |
| | Ours(25% data) | 54.06 | 79.64 | 91.16 | 0.143 | 0.552 | 0.049 | 0.025 | 0.061 |
| | Ours(50% data) | 62.57 | 86.96 | 95.39 | 0.137 | 0.521 | 0.044 | 0.025 | 0.055 |
| | Ours(75% data) | 70.55 | 90.84 | 96.21 | 0.136 | 0.518 | 0.043 | 0.025 | 0.051 |
| | **Ours** | **80.09** | **96.75** | **98.64** | **0.112** | **0.326** | **0.039** | **0.018** | **0.046** |
| REAL275 [61] | Unidepth (FT) | 33.81 | 61.97 | 88.08 | 0.152 | 0.118 | 0.134 | 0.036 | 0.138 |
| | Depth-Anything v2 (FT) | 29.87 | 44.82 | 66.09 | 0.288 | 0.187 | 0.201 | 0.092 | 0.216 |
| | Ours(25% data) | 28.64 | 53.12 | 86.23 | 0.184 | 0.127 | 0.142 | 0.036 | 0.138 |
| | Ours(50% data) | 32.75 | 56.96 | 87.83 | 0.175 | 0.109 | 0.108 | 0.029 | 0.122 |
| | Ours(75% data) | 36.50 | 64.83 | 88.77 | 0.138 | 0.097 | 0.103 | 0.025 | 0.103 |
| | **Ours** | **44.28** | **79.18** | **90.62** | **0.107** | **0.085** | **0.082** | **0.022** | **0.091** |
| ClearPose [13] | Unidepth (FT) | 12.73 | 27.36 | 40.18 | 0.175 | 0.249 | 0.118 | 0.167 | 0.247 |
| | Depth-Anything v2 (FT) | 31.23 | 56.95 | 82.21 | 0.133 | 0.076 | 0.092 | 0.155 | 0.101 |
| | Ours(25% data) | 29.59 | 53.12 | 82.02 | 0.132 | 0.075 | 0.087 | 0.159 | 0.071 |
| | Ours(50% data) | 42.69 | 63.75 | 86.55 | 0.118 | 0.071 | 0.080 | 0.151 | 0.066 |
| | Ours(75% data) | 49.92 | 70.61 | 92.76 | 0.106 | 0.062 | 0.073 | 0.138 | 0.060 |
| | **Ours** | **54.30** | **79.65** | **96.87** | **0.103** | **0.064** | **0.066** | **0.129** | **0.059** |

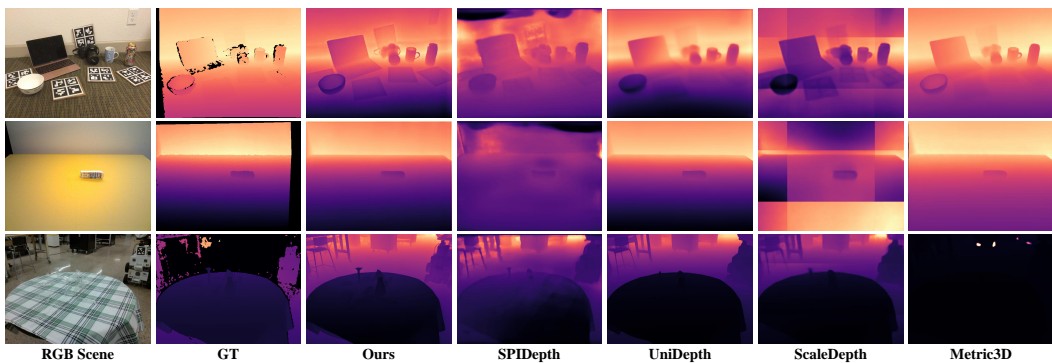

**RGB Scene**  **GT**  **Ours**  **SPIDepth**  **UniDepth**  **ScaleDepth**  **Metric3D**

Figure 4: Visualized depth prediction of ours and other metric depth estimation models: SPID-peth [53], UniDepth [49], ScaleDepth [50], and Metric3D [63]. Ours performs more clearly than other baselines, preserving all valid pixels compared to the ground truth, which misses key values.

# 4 Experiment

## 4.1 Data Preparation and Experimental Setup

We evaluate our approach on multiple 6D pose estimation datasets, each selected for its unique challenges. REAL275 [61] is chosen for its complex scenes with various objects. Toyota-Light [62], which is included in the BOP [62] challenge suite, provides a standardized testbed for evaluating robustness under challenging lighting conditions.To validate our method on transparent objects, we include ClearPose [13] and downsample the whole training set with a step size of 100. During fine-tuning, we freeze parameters in DPAv2 [26] and LoFTR [27] modules, training only our token scaler. Additional details on preprocessing and experimental setup can be found in the supplementary material (Sections D.1 and D.2).

## 4.2 Baselines and Metrics

For monocular metric depth prediction, two state-of-the-art baselines, UniDepthv1 [49] and DPAv2 [26], are selected for their strong performance and widespread use. To evaluate metric depth quality, we choose various key metrics: absolute relative error (Abs.Rel.) and square relative error (Sq.Rel.) to measure prediction errors. The threshold accuracy ($\delta_x$) gauges the proportion of predictions within $\max(\mathbf{D}/\hat{\mathbf{D}}, \hat{\mathbf{D}}/\mathbf{D}) < x$. The root mean squared error (RMSE) and mean absolute error (MAE) are selected to judge overall quality. We also include MAE in log space ($log_{10}$) to

Table 3: 6D pose estimation results on Real275 and Tyo-L dataset. We highlight our method with the `colored font` . The best and the second best values of each column are reported with **bold** and underlined fonts, respectively.

| Matcher | Depth | REAL275 Dataset | | | | | Tyo-L Dataset | | | | |
|---|---|---|---|---|---|---|---|---|---|---|---|
| | | AR ↑ | VSD ↑ | MSSD ↑ | MSPD ↑ | ADD ↑ | AR ↑ | VSD ↑ | MSSD ↑ | MSPD ↑ | ADD ↑ |
| SIFT [64] | Oracle | 34.1 | 16.5 | 37.9 | 48.0 | 16.4 | 30.3 | 7.3 | 39.6 | 44.1 | 14.1 |
| | DPAV2 | 2.1 | 1.2 | 1.5 | 3.5 | 1.3 | 5.2 | 1.3 | 3.6 | 10.6 | 3.2 |
| | Ours | 12.2 | 4.9 | 12.2 | 19.7 | 5.1 | 20.5 | 5.6 | 22.1 | 33.9 | 11.0 |
| Oryon [6] | Oracle | 46.5 | 32.1 | 50.9 | 56.7 | 34.9 | 34.1 | 13.9 | 42.9 | 45.5 | 22.9 |
| | DPAV2 | 3.7 | 1.6 | 4.0 | 5.5 | 1.5 | 6.0 | 2.5 | 4.1 | 11.3 | 3.5 |
| | Ours | 20.4 | 6.9 | 22.3 | 32.1 | 10.1 | 24.1 | 7.8 | 24.7 | 39.9 | 11.7 |
| RoMA [65] | Oracle | 41.7 | 28.6 | 44.9 | 53.8 | 30.0 | 36.8 | 19.6 | 44.7 | 46.1 | 24.5 |
| | DPAV2 | 3.7 | 1.6 | 4.1 | 5.4 | 1.2 | 5.9 | 3.2 | 3.3 | 11.2 | 3.8 |
| | Ours | 19.8 | 6.5 | 21.6 | 31.4 | 9.3 | 30.9 | 13.7 | 32.8 | 46.2 | 13.0 |
| Ours | Oracle | 56.8 | 41.2 | 63.0 | 66.2 | 56.9 | 42.0 | 34.2 | 44.1 | 47.6 | 35.0 |
| | DPAV2 | 4.1 | 2.1 | 3.5 | 6.1 | 1.4 | 5.8 | 2.8 | 3.5 | 11.1 | 2.9 |
| | Ours | **28.7** | **7.8** | **29.9** | **45.4** | **11.6** | **31.7** | **13.9** | **33.8** | **47.3** | **13.1** |
| vs. Oryon [6] | Δ | +8.3 | +0.9 | +10.6 | +13.3 | +1.5 | +7.6 | +6.1 | +9.1 | +7.4 | +1.4 |

better capture depth variations in distant objects. For 6D pose estimation, we adopt SIFT [64] and Oryon [6] as reference-based baselines. We report Average Recall (AR) across VSD, MSSD, and MSPD [62] and compute ADD(S)-0.1d to assess 3D position error. Section D.3 provides detailed mathematical definitions of each metric.

## 4.3 Quantitative Results

Table 2 presents the comprehensive results of depth prediction, and the visual comparison with more models is illustrated in Figure 4. Our method demonstrates strong performance. Using only half of the fine-tuning data, we match Unidepth's [49] performance. Naturally, performance improves with better comprehension of the scene scale when more training data is available. With all fine-tuning data, we achieve a 65% improvement in $\delta_{1.05}$ over DPAv2 [26] in Tyo-L [62]. In REAL275 [61] and ClearPose [13], we outperform DPAv2 [26] by 14.41% and 23.07%, respectively. Table 3 and 4 show the performance of pose estimation across the three benchmarks. Our method surpasses both SIFT [64] and Oryon [6], achieving average AR improvements of +15.3% and +6.5% with ground truth depth and +12.6% and +6.1% with predicted depth, respectively. This results from our depth-aware matching, which better utilizes spatial information. Compared to DPAv2 [26] (with fine-tuned head), our depth prediction produces substantial improvements: +14. 4% in accuracy with Oryon [6] matching and +20.3% with our LoFTR-based matching approach. This improvement benefits from the proposed losses, which enhance the reconstruction of geometric characteristics.

Table 4: 6D pose estimation results on ClearPose. We highlight our method with the `colored font` . The best and the second best values of each column are reported with **bold** and underlined fonts, respectively.

| Matcher | Depth | AR ↑ | VSD ↑ | MSSD ↑ | MSPD ↑ | ADD ↑ |
|---|---|---|---|---|---|---|
| SIFT [64] | Manual | 19.9 | 4.8 | 18.9 | 37.1 | 16.1 |
| | Raw | 0.8 | 0.1 | 0.9 | 1.5 | 0.1 |
| | DPAV2 | 3.1 | 0.5 | 3.8 | 5.0 | 0.8 |
| | Ours | 9.2 | 1.4 | 10.0 | 16.1 | 2.2 |
| Oryon [6] | Manual | 31.1 | 10.0 | 37.9 | 45.3 | 32.2 |
| | Raw | 1.2 | 0.2 | 1.2 | 2.2 | 1.9 |
| | DPAV2 | 8.8 | 1.5 | 10.5 | 14.3 | 3.3 |
| | Ours | 17.1 | 3.8 | 18.9 | 29.6 | 7.1 |
| RoMA [65] | Manual | 32.3 | 9.3 | 39.0 | 48.5 | 33.1 |
| | Raw | 1.5 | 0.3 | 1.2 | 3.0 | 2.2 |
| | DPAV2 | 8.9 | 1.4 | 10.9 | 14.2 | 4.0 |
| | Ours | 17.7 | 4.0 | 19.1 | 29.7 | 8.6 |
| Ours | Manual | 32.4 | 10.1 | 38.8 | 48.2 | 33.4 |
| | Raw | 1.4 | 0.2 | 1.2 | 2.8 | 2.0 |
| | DPAV2 | 9.1 | 1.6 | 11.1 | 14.4 | 3.9 |
| | Ours | **19.4** | **5.1** | **19.7** | **33.5** | **10.0** |
| vs. Oryon [6] | Δ | +2.3 | +1.3 | +0.8 | +3.9 | +2.9 |

On the Toyota-Light [62] dataset, our performance is lower than Oryon [6] with DPAv2 [26] depth, as depth prediction errors affect pose estimation. Oryon [6] benefits from textual prompts and VLM, achieving slightly better results. This highlights the importance of precise depth prediction for accurate pose estimation. Figure 5 compares the 6D pose predictions of Oryon [6] and our method.

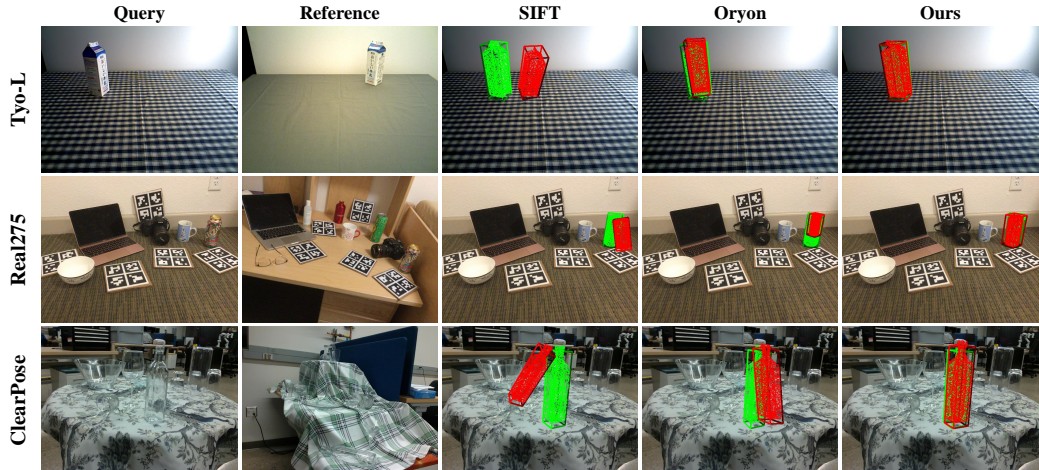

|  | Query | Reference | SIFT | Oryon | Ours |
|---|---|---|---|---|---|

Figure 5: A visual comparison of the predicted 6D pose across three datasets: the red point cloud with a 3D bounding box shows the prediction, and the green represents the ground truth. Our estimated pose exhibits less rotation error and translation shift than the baselines.

From the figure, we can tell that SIFT [64] is unable to conduct precise 6D pose, while Oryon's depth-agnostic matching [6] increases rotation errors and translation drift.

Table 5 illustrates the efficiency comparison between our methods and the baselines. Our approach reduces computational costs remarkably. This is because Oryon [6] needs a CLIP [66] to extract features, while RoMA [65] is based on high-resolution feature maps processed by ViT [67] and VGG [68].

Table 5: Comparison of the computational requirements of different matching methods.

| ID | Method | #Params. | GFLOPs | Memory(GB) |
|---|---|---|---|---|
| I | Oryon [6] | 264.3M | 120.1 | 5.90 |
| II | RoMA [65] | 111.3M | 111.8 | 5.10 |
| III | **Ours** [27] | 11.6M | 13.9 | 0.74 |

Table 6: Ablation study on the effectiveness of individual loss function on REAL275 dataset.

| $\mathcal{L}_{scale}$ | $\mathcal{L}_{edge}$ | $\mathcal{L}_{norm}$ | $\delta_{1.05}\uparrow$ | Abs.Rel.$\downarrow$ | RMSE$\downarrow$ | $\Delta(\delta_{1.05})$ |
|---|---|---|---|---|---|---|
| - | - | - | 31.16 | 0.279 | 0.281 | -13.12 |
| - | - | ✓ | 34.90 | 0.196 | 0.223 | -7.38 |
| - | ✓ | - | 35.18 | 0.188 | 0.215 | -9.10 |
| - | ✓ | ✓ | 40.23 | 0.139 | 0.162 | -4.05 |
| ✓ | - | - | 35.14 | 0.191 | 0.220 | -9.14 |
| ✓ | - | ✓ | 40.25 | 0.131 | 0.155 | -4.03 |
| ✓ | ✓ | - | 40.41 | 0.124 | 0.140 | -3.87 |
| ✓ | ✓ | ✓ | **44.28** | **0.082** | **0.107** | - |

Table 7: Ablation study on the different fine-tuning paradigms on REAL275 dataset. Full represents all parameters in the depth head that will be updated.

| FT Part | FT Type | $\delta_{1.05}\uparrow$ | Abs.Rel.$\downarrow$ | RMSE$\downarrow$ |
|---|---|---|---|---|
| *head* | LoRA [69] | 24.91 | 0.318 | 0.371 |
| *head* | Full | 29.87 | 0.201 | 0.288 |
| *head*+scaler | LoRA [69] | 43.97 | 0.091 | 0.117 |
| *head*+scaler | Full | **44.37** | **0.080** | **0.102** |
| scaler | N.A. | 44.28 | 0.082 | 0.107 |

## 4.4 Ablation Study

Table 6 shows the effectiveness of our loss components through ablation studies. Removing all three losses ($\mathcal{L}_{scale}$, $\mathcal{L}_{edge}$, and $\mathcal{L}_{norm}$) leads to a substantial drop of -13.12% in $\delta_{1.05}$ performance, demonstrating the importance of our composite loss function for accurate geometric estimation. Table 7 presents the performance across various fine-tuning setups. Fine-tuning only the depth head of DPAv2 [26], whether updating all parameters or using LoRA [69], yields poor results. Integrating our token scaler into DPAv2 [26] markedly increases performance, proving the feasibility of our method. Considering the trade-off between the extra training burden and the marginal improvement, we chose to keep the depth head frozen.

Table 8 compares our depth-aware matching against standard approaches with Oryon [6], RoMA [65], and our LoFTR [27]-based matcher. The second column determines how depth information is integrated with RGB information. In the absence of latent fusion, RGB-based and depth-based (if any) matching are performed independently, with the final correspondence obtained through the intersection of their respective results. Our method improves AR by +4.5% through

Table 8: Ablation study on the effectiveness of individual matching strategies on the Tyo-L dataset.

| Use depth | Latent fusion | Matcher | AR↑ | ADD↑ | Δ(mean) |
|---|---|---|---|---|---|
| - | - | RoMA [65] | 19.4 | 6.7 | -9.35 |
| ✓ | - | RoMA [65] | 20.8 | 7.2 | -8.40 |
| ✓ | ✓ | RoMA [65] | 30.6 | 13.0 | -0.60 |
| - | - | Oryon [6] | 24.1 | 11.7 | -4.50 |
| ✓ | - | Oryon [6] | 23.6 | 10.9 | -5.15 |
| ✓ | ✓ | Oryon [6] | 29.7 | 12.6 | -1.25 |
| - | - | LoFTR [27] | 27.2 | 11.4 | -3.10 |
| ✓ | - | LoFTR [27] | 26.6 | 11.4 | -3.40 |
| ✓ | ✓ | LoFTR [27] | **31.7** | **13.1** | - |

Table 9: Ablation study of different depth fusion mechanisms for matching (**Up**), effectiveness of token scaler and depth-aware matching (**Down**) on the Tyo-L dataset.

| Coarse Feature | Fine Feature | AR↑ | ADD↑ | Δ(mean) |
|---|---|---|---|---|
| PE [70] | PE [70] | 30.8 | 12.1 | -0.85 |
| PE [70] | Additive | 31.3 | 12.8 | -0.35 |
| Additive | PE [70] | 31.1 | 12.5 | -0.60 |
| Additive | Additive | **31.7** | **13.1** | - |

| Token Scaler | Depth-Aware Matching | AR↑ | ADD↑ | Δ(mean) |
|---|---|---|---|---|
| - | - | 4.6 | 2.0 | -19.1 |
| - | ✓ | 5.4 | 2.6 | -18.4 |
| ✓ | - | 27.2 | 11.4 | -3.1 |
| ✓ | ✓ | **31.7** | **13.1** | - |

latent fusion, which effectively incorporates depth as spatial cues to improve matching precision. Table 9 compares different feature fusion methods for depth-aware matching. We evaluated depth features as positional encoding (PE) signals [70] for LoFTR [27] transformer layers and through direct latent space addition. Our experiments reveal that simple additive fusion outperforms the PE approach while being computationally more efficient. Table 9 reports the effectiveness of each module. Without the robust depth prediction, the projected 3D point clouds are unsatisfactory, leading to a decline in performance. We also investigate the influence of other factors, such as the viewpoint gap between query and reference, and the weights of each loss term. The detailed results are illustrated in the appendix.

# 5 Conclusion

We present a novel approach for 6D pose estimation that requires only a single RGB reference image. We innovatively adapt DPAv2 [26] by fine-tuning it with a token scaler to predict metric depth, eschewing unreliable LiDAR-based depth data. We neither render multiple novel views nor build object-specific 3D templates, perform sparse reconstruction with SfM, or NeRF [25]. Instead, we directly predict a dense depth map from a single image using a frozen depth model (with lightweight tunable components), and fuse features in the 2D space. Integrating depth information into our matching pipeline substantially reduces mismatches and improves correspondence density in low-texture regions, leading to more accurate pose estimation. As our method does not rely on synthesized views or neural fields, it has a notable generalization ability in different environments. Extensive experimental results demonstrate the effectiveness and versatility of our method in various scenarios.

**Limitations**. Our approach uses object masks to localize targets and constrain correspondence matching, limiting its applicability to scenarios with available segmentation masks. Additionally, its generalization is bounded by DPAv2 [26] and pre-trained matching networks such as LoFTR [26]. Failures may occur in extremely dark conditions, where the RGB camera captures little meaningful information.

**Future Work**. Our token scaler can be utilized to fine-tune other ViT-based models. Furthermore, our depth-aware matching potentially enhances applications like scene reconstruction by providing geometric priors for multi-view images. Additionally, integrating VLMs for object localization can enhance accessibility and efficiency, ensuring a smoother user experience for a wider audience.

# 6 Acknowledgement

This research is supported by the National University of Singapore under the NUS College of Design and Engineering Industry-focused Ring-Fenced PhD Scholarship programme. This research is supported by the National Research Foundation (NRF) "Centre for Advanced Robotics Technology Innovation (CARTIN)", and also supported by the National Robotics Programme (NRP) 2.0 funding initiative "Domain-specific Robotics Foundation Models for Manufacturing (DS-RFM). The authors would like to acknowledge useful discussions with Dr.Bruce Engelmann from Hexagon, Manufacturing Intelligence Division, Simufact Engineering GmbH.

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

# Appendices of *SingRef6D: Monocular Novel Object Pose Estimation with a Single RGB Reference*

## A Overview

The supplementary material starts here. Section B illustrates the hyperparameters we used for training and inference. In Section C, some clarifications are illustrated in detail for definitions and calculations in Section 3. In Section D, we added more experimental details and clearly defined each metric mathematically. At last, we visualize the influence of each layer's weight.

## B Hyper-parameter Setting

In this section, we illustrate the detailed hyper-parameter settings for our fine-tuning process in Table 10.

Table 10: Hyperparameter Settings based on the Tyo-L dataset.

| Hyperparameter | Value |
|---|---|
| **General** | |
| Epochs | 80 |
| RGB Resolution | $640 \times 480$ |
| Learning Rate | 0.0005 |
| Warmup Epochs | 3 |
| Weight Decay | 0.001 |
| Optimizer | Adamw |
| LR Scheduler | Cosine Annealing |
| **Data** | |
| $\mathbf{D}_{min}$,$\mathbf{D}_{max}$ | 0.3,8.0 |
| **Depth Estimator** | |
| Depth-Anything Depth Scale | 19.0 |
| Efficient Attention Head | 4 |
| Efficient Attention Hidden Dim | 128 |
| Inception Conv Branch Ratio | 0.125 |
| Token Dimension | 256 |
| Batch Size | 16 |
| **LoFTR Matcher** | |
| Backbone | ResNet |
| Hidden Dims | 256 |
| $\theta_c$ | 0.2 |
| Attention Head Number | 8 |
| $\tau$ | 0.1 |
| **Loss Functions** | |
| $\mathcal{L}_{scale}$ Weight | 1.0 |
| # $\eta$ in $\mathcal{L}_{scale}$ | 0.2 |
| $\mathcal{L}_{edge}$ Weight | 0.7 |
| # $\sigma$ in $\mathcal{L}_{edge}$ | 1.0 |
| $\mathcal{L}_{norm}$ Weight | 0.6 |
| $\mathcal{L}_{reg}$ Weight | 0.5 |
| # $\alpha$ for BerHu loss | 0.9 |

## C Supplementation of Method

### C.1 Details of Depth Estimation

Depth maps are crucial for extracting 3D information, especially for 6D pose estimation with a single RGB reference. However, sensor-based depth maps often contain invalid pixels, particularly

when capturing transparent or black objects. Although human refinement of these maps can address such issues, it requires significant investment in time and labor. To overcome these challenges, we propose a novel pipeline that leverages a pre-trained DPAv2 [26], harnessing its superior generalization capabilities and scene understanding to generate high-quality depth maps without manual intervention. To enhance metric depth prediction accuracy, we augment the pre-trained model with learnable token scalers, which are fine-tuned using a curated set of labeled data, resulting in significant performance improvements.

In DPAv2 [26], for an RGB image $\mathbf{I} \in \mathbb{R}^{H \times W \times 3}$. It uses pre-trained DINOv2 [57] to extract multiscale features denoted as $\{\mathbf{F}_i \in \mathbb{R}^{H_i \times W_i \times C}\}_{i=1}^k$. $\mathbf{F}_i$ is the feature output from the $i$-th stage. To obtain the feature map containing high-level semantic and low-level details, DPAv2 [26] uses top-to-bottom fusion to integrate the features. Specifically, a trainable fusion network $\mathcal{F}_i, i \in [1, k]$ is adopted. Mathematically:

$$\mathbf{F}'_i = \mathcal{F}_i(\mathbf{F}_i, \mathbf{F}'_{i+1}), \tag{6}$$

where $\mathbf{F}'_i$ is the fused feature. Inside the fusion network, it interpolates the higher-level feature to the size of the lower-level feature and then uses a convolution network to fuse two feature maps. For $\mathbf{F}'_k$, since there is no $\mathbf{F}'_{k+1}$, the network will interpolate it to the size of $\mathbf{F}_{k-1}$ which can be treated as an upsampling. After fusing the features all the way down to $\mathbf{F}_1$, the final logits $\hat{\mathbf{D}} \in \mathbb{R}^{H \times W}$ is obtained with a conv-based depth-head:

$$\hat{\mathbf{D}} = head(\mathbf{F}'_1), \tag{7}$$

For our proposed method, the fusion process can be expressed as:

$$\mathbf{F}'_i = \mathcal{F}_i\left(\mathbf{F}_i, Scaler(\mathbf{F}'_{i+1})\right). \tag{8}$$

Moreover, we introduce a dynamic scale layer that predicts a scalar using $\mathbf{F}'_1$ to $\mathbf{F}4'$ and a trainable linear layer.

$$s_a = Linear\left(\sum_{i=1}^{4} \frac{\gamma_i}{H \cdot W} \sum_{h=0}^{H-1} \sum_{w=0}^{W-1} \mathbf{F}'_{i,h,w}\right), \tag{9}$$

where $Linear(\cdot)$ is a single linear layer, $\gamma_i$ is a learnable parameter. Therefore, the estimated depth $\hat{\mathbf{D}}$ is the prediction from the depth head multiplied by $s_a$.

Table 11 presents the impact of incorporating the dynamic scale prediction layer on model performance. Notably, this layer provides an additional boost to our models accuracy. Specifically, while the token scaler elevates $\delta_{1.05}$ from 14.64% to 72.77% compared to DPAv2 [26], the inclusion of the dynamic scale

Table 11: Performance on whether using the dynamic scale layer on the Tyo-L dataset.

| ID | $s_a$ | input | $\delta_{1.05} \uparrow$ | Abs.Rel.$\downarrow$ | RMSE$\downarrow$ |
|---|---|---|---|---|---|
| I | - | - | 72.77 | 0.051 | 0.148 |
| II | ✓ | $\mathbf{F}'_1$ | 75.43 | 0.042 | 0.142 |
| III | ✓ | Mean($[\mathbf{F}'_1, \mathbf{F}'_4]$) | 80.09 | 0.039 | 0.112 |

layer further enhances performance by an additional 7.32%. To isolate its effect, we also evaluate a framework that employs only the dynamic scale layer, omitting our token scaler. In this case, the improvement over DPAv2 [26] is limited to just 10.3%.

Alternatively, the raw depth map $\mathbf{D}_{raw}$ can serve as an optional input to our model. This approach stems from the idea that, despite potentially losing fine details, the raw depth map retains an acceptable overall sense of scale. By incorporating it, we can refine the global scale of our prediction more effectively. This process is discussed in Section C.2.

## C.2 Optional Depth Prior

In practice, the raw depth is usually available, although it might be inconsistent and inaccurate. Our pipeline can seamlessly take the raw depth as an additional input. We assume that the overall scale captured by the depth sensor is acceptable. Therefore, if the depth prior is given, we can obtain the true minimum and maximum depth values from it. Thus, refining our prediction with this "pseudo

ground truth" to obtain $D_s$:

$$\mathbf{D}_s = \left[ \left( \frac{norm(\hat{\mathbf{D}})}{min(\mathbf{D}_{raw})} - \frac{norm(\hat{\mathbf{D}})}{max(\mathbf{D}_{raw})} \right) + \frac{1}{max(\mathbf{D}_{raw})} \right]^{-1}, \tag{10}$$

where $norm(\hat{\mathbf{D}})$ is the normalized depth map to $[0,1]$. Note that in many raw depth maps, the minimal value is 0 due to depth deficiency. Therefore, we only consider the valid depth pixels whose raw depth is larger than 0.

## C.3 Details of Global Loss

In this section, we conduct a comprehensive analysis of existing loss functions to inform our global loss design, examining their relative strengths and limitations. The SSI loss in MiDas [47] is defined as:

$$\mathcal{L}_{ssi} = \frac{1}{2H \cdot W} \sum_{i=1}^{H \cdot W} \mathrm{MSE} \left( s \times \hat{d}_i + t - d_i \right), \tag{11}$$

where $\mathrm{MSE}(\cdot)$ is the mean square error, $s, t$ are predicted global scale and shift. Since our final goal is to minimize the difference between $\hat{\mathbf{D}}$ and $\mathbf{D}$ we can optimize the $s, t$ in a closed-form:

$$(s,t) = \arg\min_{s,t} \sum_{i=1}^{H \cdot W} \left( s\hat{d}_i + t - d_i \right)^2. \tag{12}$$

By defining $\mathcal{D} = [\mathbf{D}, 1], \hat{\mathcal{D}} = [\hat{D}, 1]$, and $\mathbf{g} = [s, t]$, the optimal $\mathbf{g}$ can be obtain through solving the least-square problem in Eq. 12. The optimized vector $\mathbf{g}^{opt}$ is:

$$\mathbf{g}^{opt} = \left( \sum_{i=1}^{H \cdot W} \hat{\mathcal{D}}_i \hat{\mathcal{D}}_i^\top \right)^{-1} \left( \sum_{i=1}^{H \cdot W} \hat{\mathcal{D}}_i \mathcal{D}_i \right). \tag{13}$$

Alternatively, we can use Mean Absolute Error (MAE) to replace the MSE for more robust performance on outliers. Besides, the prediction and the ground truth depth map are then normalized through:

$$\mathbf{D} = \frac{\mathbf{D} - \mathrm{median}(\mathbf{D})}{\frac{1}{M} \sum_{i=1}^{M} |\mathbf{D} - \mathrm{median}(\mathbf{D})|}, \tag{14}$$

where $M$ is the pixel number in the valid mask. Although this is more tolerant when facing more outlines, the normalization operation destroys the natural metric depth output; therefore, the MSE version of the SSI Loss is preferred for metric depth estimation. The SiLogLoss, unlike SSI Loss, supervises the overall scale in log space.

$$\mathcal{L}_{\mathrm{silog}} = \min_s \frac{1}{2M} \sum_{i=1}^{M} \left( \log \left( e^s \hat{d}_i^{-1} \right) - \log \left( d_i^{-1} \right) \right)^2. \tag{15}$$

However, the SiLogLoss ignores the potential unknown global shift where SSI Loss is not absent. Therefore, it is not selected in our pipeline.

We additionally use a gradient matching term [47] to match the depth change between prediction and ground truth.

$$\mathcal{L}_{reg} = \frac{1}{M} \sum_{k=1}^{K} \sum_{i=1}^{M} \left( \left| \nabla_x (\hat{d}_i - d_i)^k \right| + \left| \nabla_y (\hat{d}_i - d_i)^k \right| \right), \tag{16}$$

where $(\hat{d}_i - d_i)^k$ denotes the difference of disparity maps at scale $k$. We use $K = 4$ scale levels, halving the image resolution at each level.

For the BerHu term [60], it is calculated with:

$$\mathrm{BerHu}(\mathbf{D}, \hat{\mathbf{D}}) = \begin{cases} |\mathbf{D} - \hat{\mathbf{D}}|, & \text{if } |\mathbf{D} - \hat{\mathbf{D}}| \le c \\ \frac{(\mathbf{D} - \hat{\mathbf{D}})^2 + c^2}{2c}, & \text{if } |\mathbf{D} - \hat{\mathbf{D}}| > c \end{cases} \tag{17}$$

where $c$ is a threshold and set to 0.1 multiplied by the max value of $|\mathbf{D} - \hat{\mathbf{D}}|$ in our experiments.

## C.4 Details of Normal Consistency Loss

As mentioned in Section 3, the normal consistency loss focuses on optimizing the surface representation in the predicted depth map. Since the camera intrinsic is available, the projected point cloud $\mathbf{P}$ can be obtained from the corresponding depth $\mathbf{D}$. Then, we can calculate the difference vector of adjacent points, i.e., calculate the three-dimensional position difference corresponding to the adjacent horizontal and vertical pixels of each point:

$$\vec{P}_u = \vec{P}_{u+1,v} - \vec{P}_{u,v},$$
$$\vec{P}_v = \vec{P}_{u,v+1} - \vec{P}_{u,v},$$

where $\vec{P}_{u,v} = [X, Y, Z]^T$ is the three-dimensional point corresponding to pixel $(u, v)$. The normal vector can then be computed with the cross product:

$$\vec{n} = \vec{P}_u \times \vec{P}_v.$$

The weight $w_i$ aims to reduce potential outliers due to edge, occlusion, or distortion and to ensure a smooth, consistent surface representation. Therefore, it is calculated as: $\exp\left(-\lambda \|\nabla D_i\|\right)$. This exponential-based format ensures lower values in regions with large local depth gradients (typically representing object boundaries) and higher values in areas with small local depth gradients (usually corresponding to continuous object surfaces). Figure 6 illustrates the visualized weight map. It is obvious that the weight for $\mathcal{L}_{norm}$ on the objects' edges is small, while the surfaces maintain large weights. Note that we didn't filter the map with the objects' mask. In the real implementation, only foreground objects are considered.

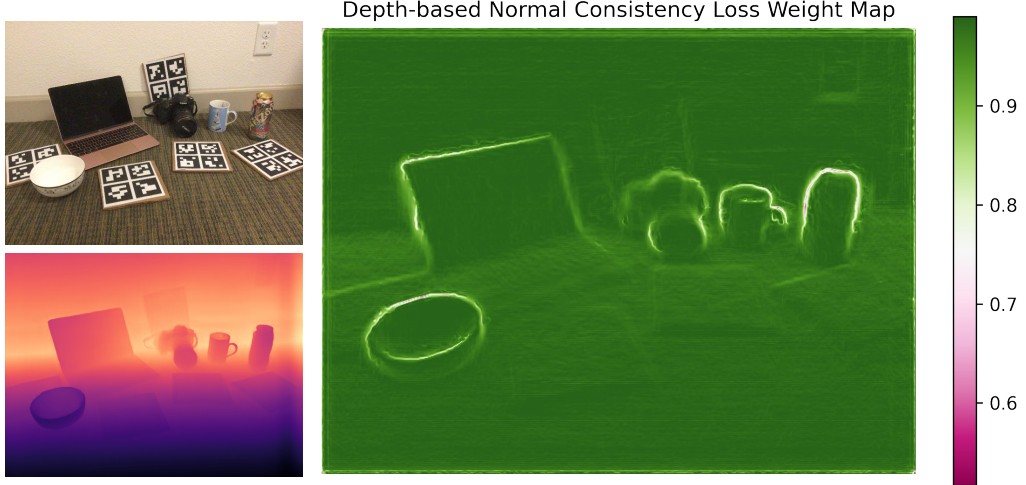

Figure 6: The RGB and the predicted depth map (**Left**) and visualized weight map for $\mathcal{L}_{norm}$ (**Right**). The green color represents a large-weight area, while the white and pink colors represent small-weight areas.

## C.5 Details of Depth-Aware Matching

Specifically, two RGB images $\mathbf{I}_q, \mathbf{I}_r$, and corresponding depth map $\mathbf{D}_q, \mathbf{D}_r$ are provided. They are first extracted by an image encoder simultaneously for feature representation with two output scales (one is $\frac{1}{8}$ of the input resolution as a coarse feature, the other is $\frac{1}{2}$ of the input resolution as a fine feature):

$$\varphi_t^{\text{coarse}}, \varphi_t^{\text{fine}} = \text{enc}(\mathbf{I}_t) + \text{norm}(\text{enc}(\mathbf{D}_t)); t \in \{q, r\}, \tag{18}$$

where $\text{enc}(\cdot)$ represents the pre-trained encoder. Note that the input depth maps are normalized to maintain a consistent scale. To ensure that depth information complements rather than dominates the matching process, we empirically normalize the depth features so they serve as an auxiliary signal. Experimental results demonstrate that without proper normalization, depth information overwhelms the matching process, leading to suboptimal performance.

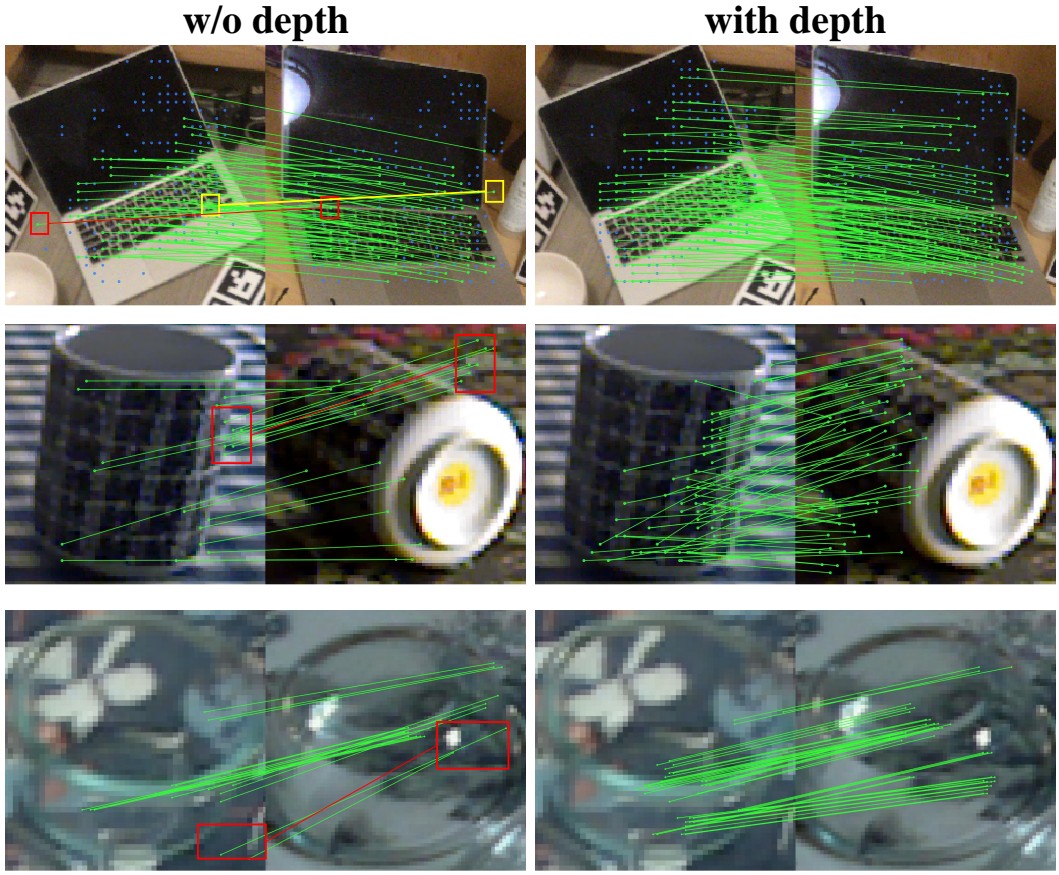

**w/o depth**           **with depth**

Figure 7: A comparison of matching results between RGB-only (**left**) and our proposed depth-aware (**right**) matching pipeline. The colored boxes with connected lines represent obvious mismatches.

The coarse feature is first processed by a transformer decoder to calculate the similarity:

$$\mathcal{S}(i,j) = \frac{1}{\tau} \cdot \langle \mathrm{dec}(\varphi_q^{\mathrm{coarse}})_i, \mathrm{dec}(\varphi_r^{\mathrm{coarse}})_j \rangle, \tag{19}$$

where $\mathrm{dec}(\cdot)$ is a transformer decoder. The matching probability is then obtained with dual-softmax [27]:

$$\mathcal{P}(i,j) = \mathrm{softmax}(\mathcal{S}(i,\cdot))_j \cdot \mathrm{softmax}(\mathcal{S}(\cdot,j))_i. \tag{20}$$

The coarse matching correspondence is then calculated with a defined threshold $\theta_c$:

$$\mathcal{M}_c = \left\{ (\tilde{i}, \tilde{j}) \mid \forall (\tilde{i}, \tilde{j}) \in \mathrm{MNN}(\mathcal{P}), \mathcal{P}(\tilde{i}, \tilde{j}) \geq \theta_c \right\}, \tag{21}$$

where $\mathrm{MNN}(\cdot)$ stands for the mutual nearest neighbor. Following the approach outlined by LoFTR [27], we extract two sets of local windows from the fine feature maps based on the coarse matches $\mathcal{M}_c$. We then compute the correlation between the center vector of $\mathrm{dec}(\varphi_q^{\mathrm{fine}})_i$ and all vectors in $\mathrm{dec}(\varphi_r^{\mathrm{fine}})_j$, generating a heatmap that represents matching probabilities between pixel $i$ and each pixel in the neighborhood of $j$. By calculating the expectation over this probability distribution, we determine the pixel position on $\mathbf{I}_r$. The aggregation of all these correspondences yields our final set of fine-level matches $\mathcal{M}_f$.

Figure 7 presents a detailed comparison between our depth-aware matching and the RGB-only matching. As shown in the figure, RGB-only matching struggles to differentiate between foreground and background pixels (indicated by yellow and red lines). This is because it primarily relies on texture and color similarity, neglecting the spatial relationship. Additionally, the density of matched points is sparse in regions with low RGB values, such as the black screen of the computer, due to the

lack of valid information in those areas. In contrast, our depth-aware matching effectively mitigates these issues, resulting in improved performance.

With such a correspondence, we can obtain the relative pose $\mathbf{T}_{q\to r}$ through deterministic point cloud registration methods [55, 71] or rigid transformation-solving algorithms [72–74]. Specifically, within the given coordinate system, the ground-truth 3-D point cloud $\mathbf{P}_g$ (under the world coordinates) can be transformed to the 3-D query point cloud $\mathbf{P}_q$ and reference point cloud $\mathbf{P}_r$ with

$$\mathbf{P}_g = \mathbf{T}_q^{-1}\mathbf{P}_q = \mathbf{T}_r^{-1}\mathbf{P}_r, \tag{22}$$

where the $\mathbf{T}_q^{-1}$ and $\mathbf{T}_r^{-1}$ are the pose matrix (from camera coordinate to world coordinate) for query and reference, respectively. As the relative pose can be used to transform the query point cloud to the reference point cloud as: $\mathbf{P}_r = \mathbf{T}_{q\to r}\mathbf{P}_q$, we can compute the 6D pose of the query object with:

$$\mathbf{T}_q^{-1} = \mathbf{T}_r^{-1}\mathbf{T}_{q\to r}. \tag{23}$$

# D    Supplementation of Experiment

## D.1    Details of Dataset Preparation

For the REAL275 [61]and Tyo-Light [62] datasets, we randomly select 80% of the separated scenes as fine-tuning data, and the rest of the scenes are processed as testing data. The ClearPose [13] dataset was constructed using sets 1, 4, 5, 6, and 7. Scene 5 from set 1 and scene 6 from sets 4-7 were designated as the test set due to their distinguished visual environments, while the remaining samples were randomly allocated to the training and testing sets. We treat the scene with a distinct background as a novel scene, although some of the objects are similar (This is because the appearance of the transparent object is heavily influenced by the background). As the original clearpose provides dense, continuous views for a scene, we use the sparse version of it, which is downsampled 100 times. The processed clearpose dataset contains 3129 images for training and 643 images for testing. For testing, we assign objects with distinct backgrounds and shapes as the targets. For each dataset, the ground truth depth maps are clamped to a given range $[\mathbf{D}_{min}, \mathbf{D}_{max}]$, and the specific value setting is illustrated in the supplementary material Section B.

Table 12: Average SSIM between fine-tuning data and test data

| Dataset | SSIM |
|---|---|
| Tyo-L | 0.2476 ($\pm$ 0.1372) |
| REAL275 | 0.1676 ($\pm$ 0.0299) |
| ClearPose | 0.1812 ($\pm$ 0.0410) |

During the experiments, we manually split the data for fine-tuning and testing to ensure the test scenes are not seen in any format by the model during fine-tuning. To quantify the similarity between test and fine-tuning data for each dataset, we use the Structural Similarity Index Measure (SSIM) [75] and then report the mean and standard deviation for each dataset.

From Table 12, we can observe that the SSIM [75] scores between the fine-tuning scenes and test scenes are relatively low, indicating limited information overlap and acceptable separation.

## D.2    Details of Experimental Setup

We fine-tune our model for 80 epochs with a batch size of 16 for all datasets except ClearPose [13], which is fine-tuned with 40 epochs for a smaller time consumption. We adopt an AdamW optimizer with an initial learning rate of 0.0005, updated by a cosine annealing scheduler with 3 warmup epochs. To avoid overfitting, we assign a weight decay of 0.01 to the optimizer. For our proposed token scaler, the hidden dimension of each layer is 256, the head number of the efficient attention layer is 4, and the intermediate dimension is 128. For the inception-conv unit, the channel number of a convolution branch is set to 32, and the intermediate dimension is set to 128 as well. All the token scalers are followed by a zero-initialized residual convolution layer, which is inspired by ControlNet [56]. The detailed hyperparameters and experiment environment settings are illustrated in the supplementary material Section B. We fine-tune the baseline models for 30 epochs using their

respective original loss functions while keeping all layers frozen except for the depth prediction heads. In Table 2, these fine-tuned variants are denoted as UniDepth (FT) [49] and Depth-Anything (FT) [26].

For the relative pose-solving part, the crop ROI size of the target object is $256 \times 256$ for efficiency. We adopt a pre-trained PointDSC [55] to solve the rigid transformation between point clouds with given correspondence points. Point correspondences within the object region were extracted using the matching result and filtered by an object mask provided by the dataset or segmentation models [54]. In REAL275 [61], we select objects from 6 distinct scenes to construct 2000 image pairs. In Tyo-L [62], we select 2000 image pairs where each query and reference image is captured under different lighting conditions. In ClearPose [13], we construct 1000 image pairs in which the query and reference images have different backgrounds, using 6 different objects. All of our experiments are conducted on an Ubuntu 22.04 server with two Nvidia RTX3090 GPUs. The deep learning framework is PyTorch 2.2.0 with CUDA 12.4.

### D.3 Details of Evaluation Metrics

The absolute relative error (Abs.Rel.)

$$\text{Abs.Rel.} = \frac{|\mathbf{D} - \hat{\mathbf{D}}|}{\mathbf{D} + \epsilon}$$

and its quadratic variant, the square relative error(Sq.Rel.)

$$\text{Sq.Rel.} = \frac{|\mathbf{D} - \hat{\mathbf{D}}|^2}{\mathbf{D} + \epsilon}$$

quantify the local deviation between predicted ($\hat{\mathbf{D}}$) and ground-truth ($\mathbf{D}$) depths. $\epsilon$ is a small value to avoid division by 0.

The root mean squared error (RMSE) and mean absolute error (MAE) are defined as:

$$\text{RMSE} = \sqrt{\frac{1}{HW} \sum_{i=1}^{H} \sum_{j=1}^{W} (\hat{d}_{ij} - d_{ij})^2}$$

$$\text{MAE} = \frac{1}{HW} \sum_{i=1}^{H} \sum_{j=1}^{W} |\hat{d}_{ij} - d_{ij}|$$

The ($\log_{10}$) metric is calculated with:

$$\log_{10} = \frac{1}{HW} \sum_{i=1}^{H} \sum_{j=1}^{W} \left| \log_{10}(\hat{d}_{ij}) - \log_{10}(d_{ij}) \right|$$

For the 6D pose estimation metrics [62], Visible Surface Discrepancy (VSD) quantifies the spatial gap between an object's surfaces when comparing its true position to its estimated position.

$$\text{VSD} = \frac{1}{|V(\hat{\mathbf{P}}, \delta) \cup V(\bar{\mathbf{P}}, \delta)|} \sum_{\mathbf{p}} \left| \mathbf{D}_{\hat{\mathbf{P}}}(\mathbf{p}) - \mathbf{D}_{\bar{\mathbf{P}}}(\mathbf{p}) \right|,$$

where $\hat{\mathbf{P}}$ is the predicted pose, $\bar{\mathbf{P}}$ is the ground-truth, and $\delta$ is a distance threshold. $V(\hat{\mathbf{P}}, \delta)$ represents the set of pixels visible under pose $\hat{\mathbf{P}}$ and whose depth difference is less than $\delta$.

The greatest distance between two poses' surfaces, accounting for symmetry, is measured by the Maximum Symmetry-Aware Surface Distance (MSSD):

$$\text{MSSD}(\hat{\mathbf{P}}, \bar{\mathbf{P}}, S_M, V_M) = \min_{S \in S_M} \max_{\mathbf{x} \in V_M} \left\| \hat{\mathbf{P}} \mathbf{x} - \bar{\mathbf{P}} S \mathbf{x} \right\|,$$

where the set $S_M$ contains global symmetry transformations of the object model $M$, $V_M$ is a set of the model vertices.

The Maximum Symmetry-Aware Projection Distance (MSPD) aims to measure how visually different an object appears (measured in pixels) when it's rendered using an incorrect pose while accounting for all possible symmetries.

$$\text{MSPD} = \min_{S \in S_M} \max_{\mathbf{x} \in V_M} \left\| \text{proj}(\hat{\mathbf{P}}\mathbf{x}) - \text{proj}(\bar{\mathbf{P}}S\mathbf{x}) \right\|,$$

where $\text{proj}(\cdot)$ represents the operation of projecting a three-dimensional point onto an image plane.

ADD(S)-0.1d measures the recall in pose estimation based on error threshold: a pose is considered correctly estimated when its error is less than 10% of the object's diameter. For an asymmetric object, the error is computed as:

$$\text{ADD} = \frac{1}{|V_M|} \sum_{\mathbf{x} \in V_M} \left\| \hat{\mathbf{P}}\mathbf{x} - \bar{\mathbf{P}}\mathbf{x} \right\|,$$

where $|V_M|$ represents the size of the vertice set, $\hat{\mathbf{P}}\mathbf{x}$ and $\bar{\mathbf{P}}\mathbf{x}$ represent the position of vertice x in the estimated and true poses respectively. For a symmetric object:

$$\text{ADD-S} = \frac{1}{|V_M|} \sum_{\mathbf{x} \in V_M} \min_{\mathbf{x}' \in V_M} \left\| \hat{\mathbf{P}}\mathbf{x} - \bar{\mathbf{P}}\mathbf{x}' \right\|,$$

where $\mathbf{x}'$ represents the closest symmetry point to $\mathbf{x}$ in the 3D model.

### D.4 Sensitivity Analysis

We conduct the sensitivity analysis on the weight of $\mathcal{L}_{norm}$, $\mathcal{L}_{edge}$, and $\mathcal{L}_{scale}$. Figure 8 illustrates the curve of the results. We consolidate the final value of weights through extensive experiments.

### D.5 More Experimental Result

#### D.5.1 Cross-domain Depth Fine-tuning

To further validate the cross-domain generalization capability of our method, we conducted additional experiments involving domain transfer evaluation. Specifically, we fine-tuned the depth estimation model on a single source dataset (e.g., REAL275 [61], which primarily contains industrial and everyday objects under static lighting conditions) and directly evaluated its performance on target datasets with different characteristics, ClearPose [13] (transparent objects in complex spatial environments) and Toyota-Light [62] (dynamic lighting variations)without any domain adaptation or additional fine-tuning. These three datasets exhibit distinct scales and environmental characteristics with substantial variations in camera viewpoints and intrinsic parameters, thereby enabling comprehensive validation of our method's cross-domain generalization capabilities. The cross-domain evaluation results are presented in Table 13.

Table 13: Quantitative result of metric depth estimation with cross-validation. The first column denotes the generalization direction from the fine-tuned dataset to the evaluation datasets.

| Setting | Method | $\delta_{1.05} \uparrow$ | $\delta_{1.10} \uparrow$ | $\delta_{1.25} \uparrow$ | RMSE$\downarrow$ | $log10 \downarrow$ | Abs.Rel.$\downarrow$ | Sq.Rel.$\downarrow$ | MAE$\downarrow$ |
|---|---|---|---|---|---|---|---|---|---|
| | Unidepth (FT) | 8.35 | 22.61 | 42.01 | 0.983 | 1.419 | 1.855 | 0.702 | 0.500 |
| REAL275 $\rightarrow$ Tyo-L | DPA v2 (FT) | 10.55 | 28.93 | 50.23 | 0.519 | 0.764 | 0.399 | 0.177 | 0.326 |
| | **Ours** | **28.13** | **34.91** | **55.72** | **0.377** | **0.471** | **0.223** | **0.119** | **0.218** |
| | Unidepth (FT) | 8.74 | 14.02 | 25.91 | 0.364 | 0.268 | 0.294 | 0.312 | 0.273 |
| REAL275 $\rightarrow$ ClearPose | DPA v2 (FT) | 20.85 | 44.48 | 62.36 | 0.258 | 0.232 | 0.207 | 0.280 | 0.181 |
| | **Ours** | **42.19** | **66.25** | **79.70** | **0.182** | **0.106** | **0.131** | **0.228** | **0.109** |

The experimental results show that cross-domain performance drops as expected, reflecting the domain sensitivity inherent in depth estimation tasks. Nevertheless, our method consistently outperforms fine-tuned baselines (e.g., DPA v2 [26] and UniDepth [49]) under the same cross-domain settings. This superior generalization stems from our token scaler's design philosophy. Unlike methods tailored for specific datasets, it adaptively modulates intermediate features in a data-agnostic manner. This generic approach enables better generalization across diverse visual inputs without being constrained by particular dataset characteristics.

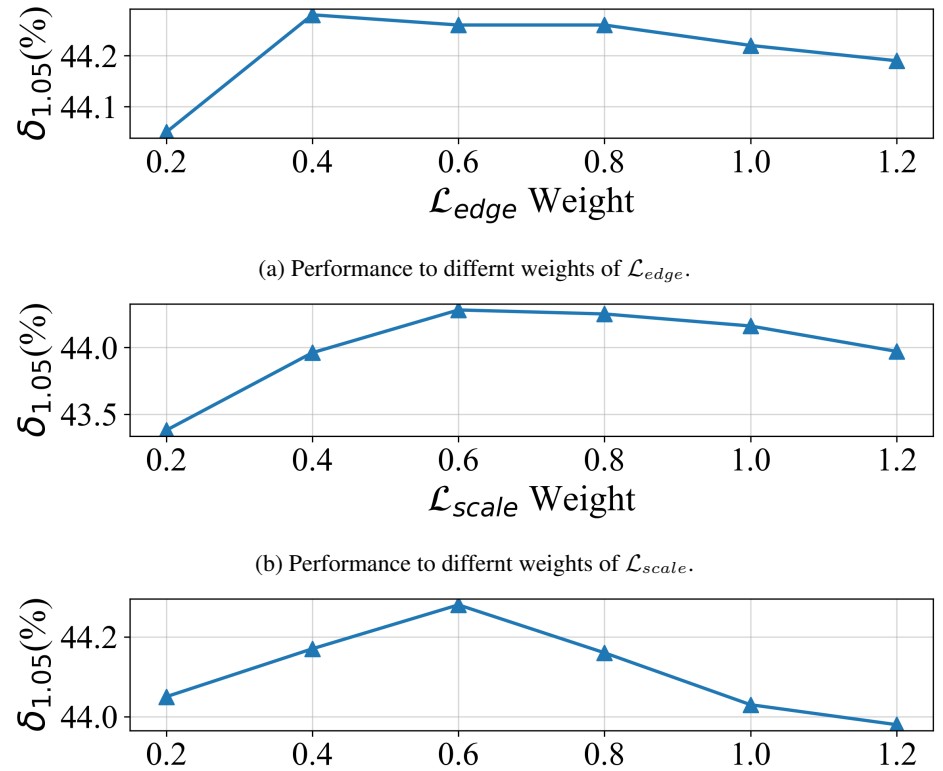

(a) Performance to differnt weights of $\mathcal{L}_{edge}$.

(b) Performance to differnt weights of $\mathcal{L}_{scale}$.

(c) Performance to differnt weights of $\mathcal{L}_{norm}$.

Figure 8: Sensitivity analysis to the weights of our proposed losses on REAL275.

Moreover, the introduced loss functions are designed to regularize the model for consistent geometric understanding. By leveraging cues such as edges and surface normals, they focus on capturing generalizable geometric structures rather than overfitting to the distribution of a particular dataset. Therefore, this suggests that the improvements brought by our approach are not solely due to domain-specific tuning but also stem from its advanced intrinsic mechanism. Besides, the proposed modules are inherently transferable and can be seamlessly integrated into other visual tasks (such as depth-based navigation, etc.), demonstrating strong potential for broader applicability beyond the current setting for other directions.

### D.5.2 Results On Additional Pose Benchmarks

We conducted two additional experiments on LM-O [62] and YCB-V [62], which are representative benchmarks featuring occlusions and weakly textured objects, such as monochromatic, single-material objects. We evaluate pose estimation on three different target objects, generating 120 image pairs for each. Table 14 reports the quantitative results.

The experimental results show that heavy occlusion still causes a significant drop in performance. However, our method consistently outperforms the baseline significantly within the single RGB reference setting. This improvement stems from our framework's human vision-inspired mechanism, which adaptively adjusts features across different layers using both appearance and spatial information. While conventional approaches like Oryon [6] or SIFT [64] rely primarily on RGB similarity, our matching strategy incorporates spatial context to improve alignment accuracy. Additionally, we maintain spatial consistency throughout both coarse and fine-grained matching stages (see Figure 1). These designs result in substantially better robustness, particularly in challenging scenarios where appearance cues alone prove inadequate, such as under heavy occlusion.

Table 14: 6D pose estimation results on YCB-V and LM-O dataset.

| Dataset | Matcher | Depth | AR ↑ | VSD ↑ | MSSD ↑ | MSPD ↑ | ADD ↑ |
|---------|---------|-------|------|-------|--------|--------|-------|
| LM-O | Oryon | Oracle | 6.1 | 3.1 | 6.9 | 8.3 | 5.5 |
| | | Ours | 3.8 | 1.9 | 4.2 | 5.4 | 2.7 |
| | Ours | Oracle | 9.9 | 4.3 | 11.5 | 13.9 | 8.7 |
| | | Ours | **5.3** | **2.6** | **5.7** | **7.3** | **3.8** |
| | vs. Oryon | Δ | +2.7 | +0.95 | +3.1 | +3.8 | +2.2 |
| YCB-V | Oryon | Oracle | 8.6 | 4.7 | 9.3 | 11.8 | 7.5 |
| | | Ours | 5.8 | 2.4 | 6.6 | 8.4 | 5.2 |
| | Ours | Oracle | 11.4 | 5.2 | 12.3 | 15.7 | 8.9 |
| | | Ours | **7.9** | **3.7** | **8.5** | **11.6** | **6.4** |
| | vs. Oryon | Δ | +2.45 | +0.90 | +2.45 | +3.55 | +1.30 |

For multi-object situations, our pipeline naturally extends to these scenarios by simply incorporating an off-the-shelf object detector or segmenter, such as SAM [54] or SAM2 [76]. This design highlights the generality and scalability of our approach, making it readily adaptable to diverse real-world applications without requiring major architectural changes. The experiment results illustrate the robustness of our method under occlusion scenarios.

### D.5.3 Comparison With Additional Baselines

We find that Any-6D [43], FS6D [10], and the original LoFTR [27] are comparable methods. Therefore, to further investigate the performance of them compared to our SingRef6D. We conduct several experiments and report the results in Table 15.

Table 15: More 6D pose estimation results on the Tyo-L dataset.

| Matcher | Depth | AR ↑ | VSD ↑ | MSSD ↑ | MSPD ↑ | ADD ↑ |
|---------|-------|------|-------|--------|--------|-------|
| FS6D [10] | Oracle | 14.1 | 5.2 | 17.9 | 19.2 | 10.0 |
| | DPAV2 | 1.6 | 0.5 | 1.9 | 2.4 | 0.3 |
| | Ours | 9.3 | 1.8 | 6.3 | 7.8 | 3.2 |
| Any-6D [43] | Oracle | 43.3 | 15.8 | 55.8 | 58.4 | 32.2 |
| | DPAV2 | 5.2 | 1.3 | 4.3 | 11.0 | 2.2 |
| | Ours | 31.6 | 13.5 | 33.5 | 47.8 | 13.2 |
| Ours | Oracle | 42.0 | 34.2 | 44.1 | 47.6 | 35.0 |
| | DPAV2 | 5.8 | 2.8 | 3.5 | 11.1 | 2.9 |
| | Ours | **31.7** | **13.9** | **33.8** | **47.3** | **13.1** |

Our method outperforms FS6D [10] across the selected evaluation metrics. Any-6D [43], benefiting from its strong 3D reconstruction backbone, achieves slightly better results than ours when using annotated depth. However, it exhibits a strong dependency on the quality of depth inputits performance drops significantly when using DPAv2-predicted depth. This is because Any-6D [43] adopts a more global optimization strategy (e.g., point cloud registration or global pose supervision), which effectively constrains the maximum deviation (results in higher recall on MSSD and MSPD). However, the accumulation of depth errors in local regions leads to lower recall of VSD. In contrast, our method emphasizes local pixel-wise alignment (e.g., fine-grained optimization based on depth maps), resulting in higher recall on VSD. Nonetheless, it is less robust in handling certain keypoints or symmetries, which leads to lower recall on MSSD and MSPD.

The performance gap between Any-6D [43] and our method with different depth inputs, and the differences seen when switching from oracle to predicted depth within each method, show that our pipeline is more robust and overall superior for both pose estimation and depth prediction. Furthermore, our method delivers more stable depth reconstruction and enhanced geometric awareness for pose estimation. Designed modularly, our methods can be integrated seamlessly into downstream pose estimation pipelines or incorporated as components within other depth estimation models. On

the Tyo-L [62] dataset, vanilla LoFTR [27] for RGB-based matching yields an AR of 27.2, while our method achieves 31.7. This demonstrates more robust and accurate feature matching of our method under identical conditions.

In addition, we conduct the comparison studies with DVMNet [46] and 3DAHV [45] by iteratively inferring from the image pair. Table 16 reports the corresponding results.

Table 16: Comparison with other models on LM-O dataset

| Method | AR | ADD | Angular Error |
|---|---|---|---|
| 3DAHV [45] | 7.7 | 6.3 | 52.71 |
| DVMNet [46] | 8.6 | 7.4 | **51.66** |
| Ours | **9.9** | **8.7** | 51.89 |

While our method exhibits slightly higher angular error compared to DVMNet [46], it achieves superior performance in terms of AR and ADD, reflecting more accurate translation estimation. This difference stems from the fact that DVMNet [46] and 3DAHV [45] are supervised with full ground-truth metric poses and trained on large-scale datasets with extensive pose annotations. In contrast, our method is only fine-tuned using predicted depth and does not require additional pose supervision.

Furthermore, DVMNet [46] and 3DAHV [45] focus on relative pose estimation under the same-scene conditions with encoded latent 3D representation, whereas our method is designed to handle cross-scene situations and estimate 6D poses. This highlights a key distinction: DVMNet [46] is suitable for applications requiring accurate $SO(3)$ rotation estimation in stable environments, while our approach provides a more generalizable and lightweight solution for full-pose estimation without dependence on dense pose labels. This advantage facilitates the implementation of our method in real-world cases.

### D.5.4 Comparison With Additional Registration Methods

Iterative Closest Point (ICP) [77] is a well-known algorithm for point registration; we admit ICP [77] is a baseline worthy of comparison. We conducted ICP [77] experiments using point clouds from both reference and query views. The results are shown in the Table 17.

Table 17: 6D Pose estimation result on the ClearPose dataset compared to ICP.

| Matcher | Depth | AR ↑ | VSD ↑ | MSSD ↑ | MSPD ↑ | ADD ↑ |
|---|---|---|---|---|---|---|
| | Manual | 19.8 | 5.5 | 23.2 | 32.9 | 20.6 |
| ICP [77] | Raw | 0.3 | 0.04 | 0.25 | 0.61 | 0.9 |
| | Ours | 8.1 | 2.6 | 9.4 | 12.4 | 4.5 |
| | Manual | 32.4 | 10.1 | 38.8 | 48.2 | 33.4 |
| Ours | Raw | 1.4 | 0.2 | 1.2 | 2.8 | 2.0 |
| | Ours | **19.4** | **5.1** | **19.7** | **33.5** | **10.0** |

ICP [77] performs unsatisfactorily due to its sensitivity to initialization and noise, making correspondence unreliable. In contrast, our depth-aware matching operates in latent space, preserving pixel-level consistency and mitigating the pitfalls of noisy 3D reconstructions. Our method is enriched with spatial cues in a coarse-to-fine manner, further boosted following pose solving, performing better under transparent and low-texture conditions where ICP struggles.

We also consider learning-based registration methods [78, 71] to substitute ICP [77]. Table 18 shows that although LePard [71] and EYOC [78] are notable 3D registration methods, their performance under the proposed setting is lower than our method. Directly applying registration methods to raw reconstructed point clouds yields suboptimal performance due to two key limitations. First, in our experimental setup, there is only 1 reference sample, which is not guaranteed to originate from the same scene as the query. This makes it challenging for registration-based approaches to infer reliable relative spatial relationships from scene geometry alone, as a single cross-scene

Table 18: 6D pose estimation results on three benchmarks compared to learning-based registration methods.

| Method | Real275 | | Tyo-L | | ClearPose | |
|---|---|---|---|---|---|---|
| | AR | ADD | AR | ADD | AR | ADD |
| LePard [71] | 11.9 | 5.1 | 14.8 | 6.5 | 8.0 | 4.2 |
| EYOC [78] | 13.1 | 5.5 | 15.2 | 7.0 | 8.3 | 4.4 |
| Ours | **28.7** | **11.6** | **31.7** | **13.1** | **19.4** | **10.0** |

RGB reference inherently provides a limited overlapped region (even using ground-truth depth). Second, point clouds derived from depth estimation are frequently incomplete due to occlusions and viewpoint variations, which significantly undermine registration effectiveness.

In contrast, our method first employs depth-aware matching to establish initial correspondences between query and reference images. These correspondences directly guide subsequent pose estimation, enabling our pipeline to reduce dependence on the completeness of reconstructed point clouds. This design allows our method to maintain robust performance even in cross-scene scenarios or when dealing with partial or noisy point cloudsa key advantage over approaches that rely exclusively on learning-based registration.

Additional experiments reveal that applying ICP [77] refinement after the initial registration leads to only marginal gains (for LePard [71]: +0.12 AR on Real275 [61], +0.06 AR on ClearPose [13], and +0.11 AR on Tyo-L [62]), with negligible differences in ADD. This underscores the role of accurate initial correspondence and further substantiates the effectiveness of our approach. Notably, despite utilizing a lightweight architecture, our method surpasses learning-based registration models under our single-RGB cross-scene reference settings, demonstrating both the practicality and superiority of the proposed framework.

### D.5.5 Results on Benchmark with Reflective Objects

To provide an analysis on reflective objects specifically, we fine-tuned our depth model on a subset of the training data and evaluated 6D pose estimation on the validation subset of HouseCat6D [79]. We manually selected three reflective objects (teapoint, knife, and cup) as targets and generated 100 image pairs for each.

Table 19: More 6D pose estimation results on the HouseCat6D dataset.

| Matcher | Depth | AR ↑ | VSD ↑ | MSSD ↑ | MSPD ↑ | ADD ↑ |
|---|---|---|---|---|---|---|
| Oryon [6] | Oracle | 57.4 | 41.0 | 62.6 | 68.7 | 68.8 |
| | DPAV2 | 26.6 | 9.1 | 33.6 | 37.3 | 39.5 |
| | Ours | 38.1 | 20.8 | 44.4 | 49.3 | 49.0 |
| Ours | Oracle | 63.2 | 48.2 | 73.0 | 78.6 | 76.2 |
| | DPAV2 | 30.1 | 12.0 | 41.9 | 46.3 | 43.8 |
| | Ours | **44.5** | **26.1** | **51.8** | **55.5** | **55.2** |
| vs. Oryon | Δ | +6.4 | +5.3 | +7.4 | +6.2 | +6.2 |

As shown in Table 19, our method outperforms the baseline and achieves higher depth quality compared to the vanilla fine-tuned DPAv2 [26]. This is because baseline methods like Oryon [6] heavily rely on RGB information, while the HouseCat6D [79] dataset is specifically designed to challenge such dependence. It provides dynamic variations in RGB cues through high-resolution images and carefully selected reflective objects, which naturally exhibit different surface appearances from varying viewpoints. This makes it difficult for RGB-dependent methods like Oryon [6] to establish reliable correspondences.

In contrast, our approach employs a coarse-to-fine matching strategy that effectively leverages depth cues. This enables robust matching even when RGB signals are unreliable, as our method can rely

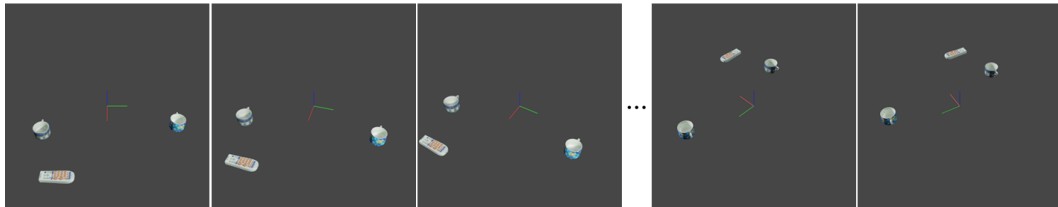

Figure 9: The rendered images obtained from different viewpoints are referenced with respect to the $Z$ axis.

on geometric structures and spatial consistency. Furthermore, the use of high-resolution feature maps allows for refined correspondence estimation, reducing errors in challenging cases. These results not only demonstrate the architectural advantage of our method but also highlight its practical applicability in real-world scenarios where RGB signals may dynamically vary.

### D.5.6  Influences of Query-Reference Viewpoint Gap

We conduct an experiment that manually rotates the scene with a specific angle $\alpha \in [15°, 150°]$ along the $X$, $Y$, and $Z$ axes, respectively. Specifically, we use accessible 3D models (e.g., from BOP [62] datasets) to construct a scene with BlenderProc and rotate the camera by $\alpha$ (The reference scene and the query differ in background and contain different objects, except for the target object). We define a right-handed coordinate system, where the $Z$-axis points vertically upward, perpendicular to the object surface, and the $X$ and $Y$ axes lie in the horizontal plane. The camera is positioned at $(1, 1, 1)$, measured in meters, looking down at the scene, with all objects placed on the $XY$ plane. Since most existing benchmarks adopt a top-down perspective, we follow the same setup. Figure 9 illustrates an example of this albation environment. To evaluate the effect of viewpoint variation, we report results under different camera rotations along the $X$ and $Z$ axes. Rotation along the $Y$ axis is omitted, as we observed a strong correlation between rotations around the $X$ and $Y$ axes during the experiments, with a Spearman coefficient exceeding 0.86. This correlation is likely due to the symmetry of the setup. Therefore, we report only the $X$-axis rotation in the Table 20 to represent horizontal-axis variations. In future revisions, we plan to include finer-grained angular distinctions and explicit values for $Y$-axis rotations on more scenes.

Table 20: Results of rotation differences between reference and query viewpoints along three axes.

| Axis\$\alpha$ | 15° | | 30° | | 45° | | 60° | | 75° | | 90° | | 120° | | 150° | |
|---|---|---|---|---|---|---|---|---|---|---|---|---|---|---|---|---|
| | AR | ADD | AR | ADD | AR | ADD | AR | ADD | AR | ADD | AR | ADD | AR | ADD | AR | ADD |
| X | 69.84 | 65.70 | 61.17 | 60.42 | 50.52 | 54.38 | 42.36 | 46.91 | 33.71 | 38.64 | 30.12 | 33.47 | 22.83 | 24.36 | 20.54 | 21.90 |
| Z | 74.26 | 72.19 | 71.50 | 69.83 | 68.13 | 67.21 | 65.80 | 64.02 | 62.42 | 60.57 | 58.31 | 56.48 | 39.76 | 36.73 | 31.95 | 28.54 |

From the table, we observe that pose estimation performance degrades with increasing viewpoint differences between the reference and query images, especially when the rotation is along the X-axis. This is because X-axis rotations significantly alter perspective and occlusion patterns, making it more challenging for the model to match spatial features. In contrast, Z-axis rotations mainly induce in-plane transformations with relatively mild geometric distortion, leading to a more gradual performance drop. However, when the rotation exceeds 90°, even Z-axis changes cause noticeable degradation. This is likely due to decreased visual discriminability  for example, the front of a mug may have a distinct texture or logo, while the back is plain, reducing the object's appearance discriminability and making it harder for the model to rely on appearance and forcing it to depend solely on geometry and spatial consistency.

Despite this challenge, our method exhibits a smaller performance drop (65%) under such extreme conditions, while baselines like Oryon [6] and SIFT [64] fail almost entirely, with performance degradation exceeding 90%. This robustness stems from our models ability to effectively exploit spatial consistency and adapt spatial representations through the proposed token scaler, thus boosting geometric understanding in the coarse-to-fine matching process. This mechanism helps to improve matching accuracy when RGB cues are unreliable. Furthermore, our loss functions impose strong geometric constraints such as normal consistency and edge emphasis, enabling the model to maintain

Table 21: Pose Estimation Result (AR) with Binned Query-Reference Viewpoint Difference

| Bin | Real275 | Tyo-L | ClearPose |
|---|---|---|---|
| $\left[0, \frac{\pi}{4}\right)$ | 33.1 | 41.5 | 31.1 |
| $\left[\frac{\pi}{4}, \frac{\pi}{2}\right)$ | 27.2 | 34.2 | 20.1 |
| $\left[\frac{\pi}{2}, \frac{3\pi}{4}\right)$ | 23.3 | 30.1 | 16.6 |
| $\left[\frac{3\pi}{4}, \pi\right]$ | 19.9 | 26.8 | 11.8 |

a reasonable estimation accuracy even under large viewpoint changes. These results and analyses demonstrate the robustness of our model and highlight its potential for real-world applications.

To further investigate the influence of the viewpoint gap statistically, we provide a quantitative breakdown of the pose estimation results on these three datasets, binned by the query-reference viewpoint difference with the geodesic distance [80]. We adopt the same definition as in [80], where geodesic distance on a unit sphere is calculated with $SO(3)$ rotations and ranges from 0 to $\pi$. Based on this, we performed a bin-wise statistical analysis on the three benchmark datasets used in our study. The ARs are shown in Table 21. Note that all statistical numbers are calculated based on ground-truth pose annotations. We conducted a basic stability check and removed a few outlier samples with abnormally low overlap, ensuring that the resulting subset maintained a similar distribution to the overall dataset. Additionally, we examined the distribution of ground-truth distances and found that the proportions across different intervals were generally consistent, with no single range overwhelmingly dominating the distribution.

The performance of pose estimation deteriorates as the geodesic distance between the query and reference increases. On the ClearPose [13] dataset, we observe that when the geodesic distance range shifts from $\left[0, \frac{\pi}{4}\right)$ to $\left[\frac{\pi}{2}, \frac{3\pi}{4}\right)$, the Average Recall (AR) of our method decreases by approximately **29.6**%, whereas AR of Oryon [6] declines by **50.3**%. This further substantiates the superiority of our approach over the baseline under the proposed evaluation setting. It is important to note that, since our query-reference pairs originate from different scenes, additional factors-such as variations in background, occlusions caused by surrounding objects, and changes in scene layout-also influence performance fluctuations. For instance, on the REAL275 [61] dataset, the AR decreases by approximately **29.6**% for the same geodesic distance shift ( $\left[0, \frac{\pi}{4}\right) \rightarrow \left[\frac{\pi}{2}, \frac{3\pi}{4}\right]$ )), compared to a **46.6**% drop on ClearPose [13], which contains glass objects. This is because glass objects in ClearPose [13] are more difficult for the model to capture common appearance features, indicating that the intrinsic material properties of objects also impact this performance trend.

### D.6 More Visualizations

In this section, we visualize the prediction from two notable metric depth estimation models, SPIdepth [53] and Unidepth [49]. Figure 4 illustrates the comparison between ours and theirs. UniDepth [49], though it estimates acceptable scales, lacks enough representation of objects' details. SPIDepth [53], while identifying the object well, outputs an unsatisfactory overall scale.

In addition, we present visualizations of depth predictions using different weight distributions across $\mathbf{F}_1$ to $\mathbf{F}_4$. The term $\frac{\text{Scaled } \mathbf{F}}{\mathbf{F}}$ represents the ratio of the mean value of the feature processed by the token scaler to that of the original feature. The rows corresponding to 0.0 represent cases where we manually initialized the output convolution layer weights to zero, resulting in the network effectively ignoring these layers during initial training iterations. Figure 10, Figure 11, and Figure 12 shows the visualization results of REAL275 [61], ClearPose [13], and Tyo-L [62] dataset, respectively. The special cases are illustrated on the upper right of each figure. "0.0 for all" means the token scaler is a zero constant, which deactivates the fusion process, and the prediction is obtained purely from the vanilla $\mathbf{F}_1$. "-1.5 for all" or "-1.0 for all" are special cases in which we manually clamp the scaler output to negative values and deploy such a strategy to all layers. It is obvious from the figure that "0.0 for all" results in low-quality output that lacks clear representation and distinguishment of objects and background. "-1.5 for all" or "-1.0 for all" leads to meaningless output since a too large weight may break the distribution within the feature map, thus impairing the inference process in the depth head.

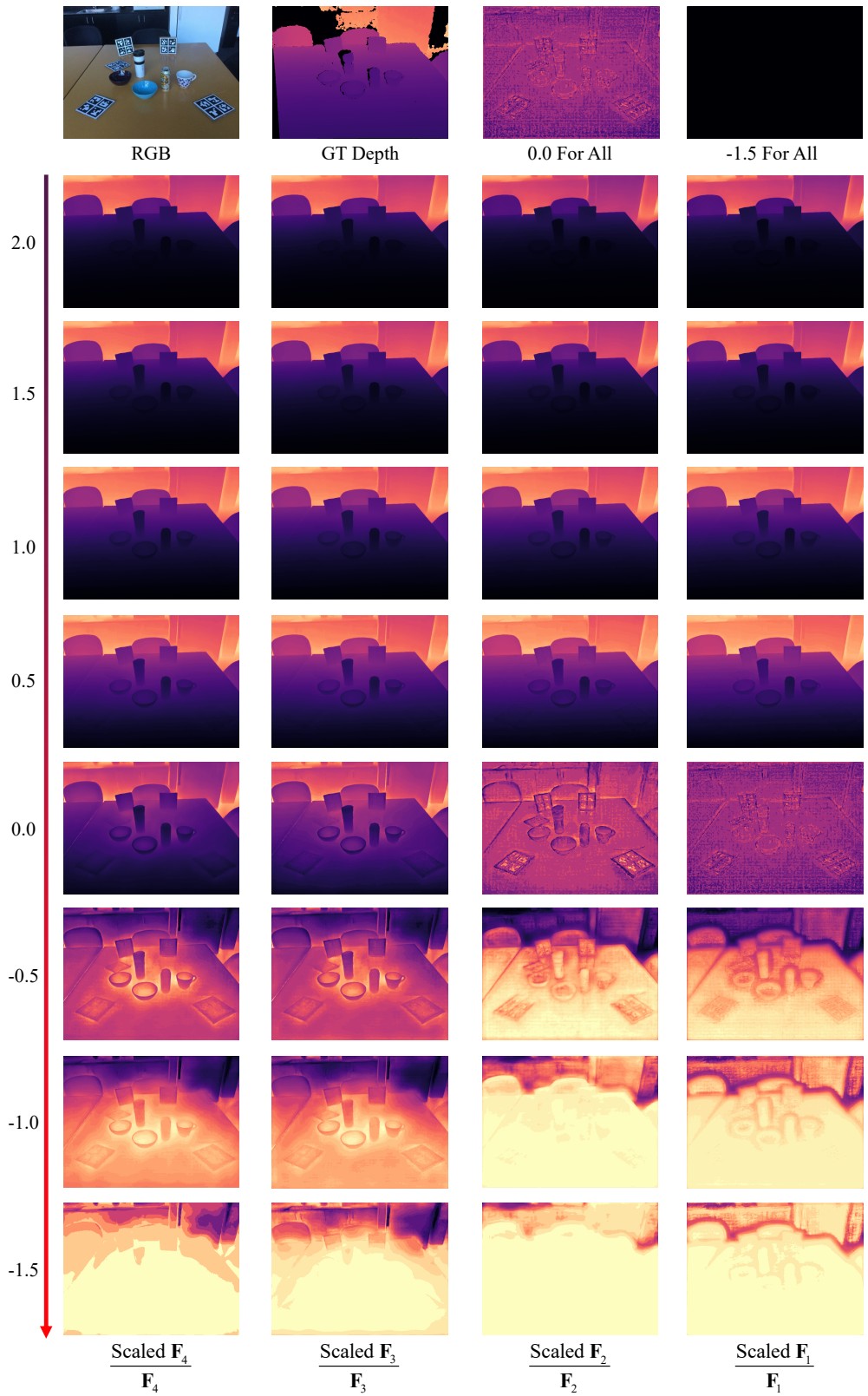

Figure 10: Comparison of different fusion weights on Real275 dataset. Each column is independent and varies the corresponding weight from 1.0 to -2.5.

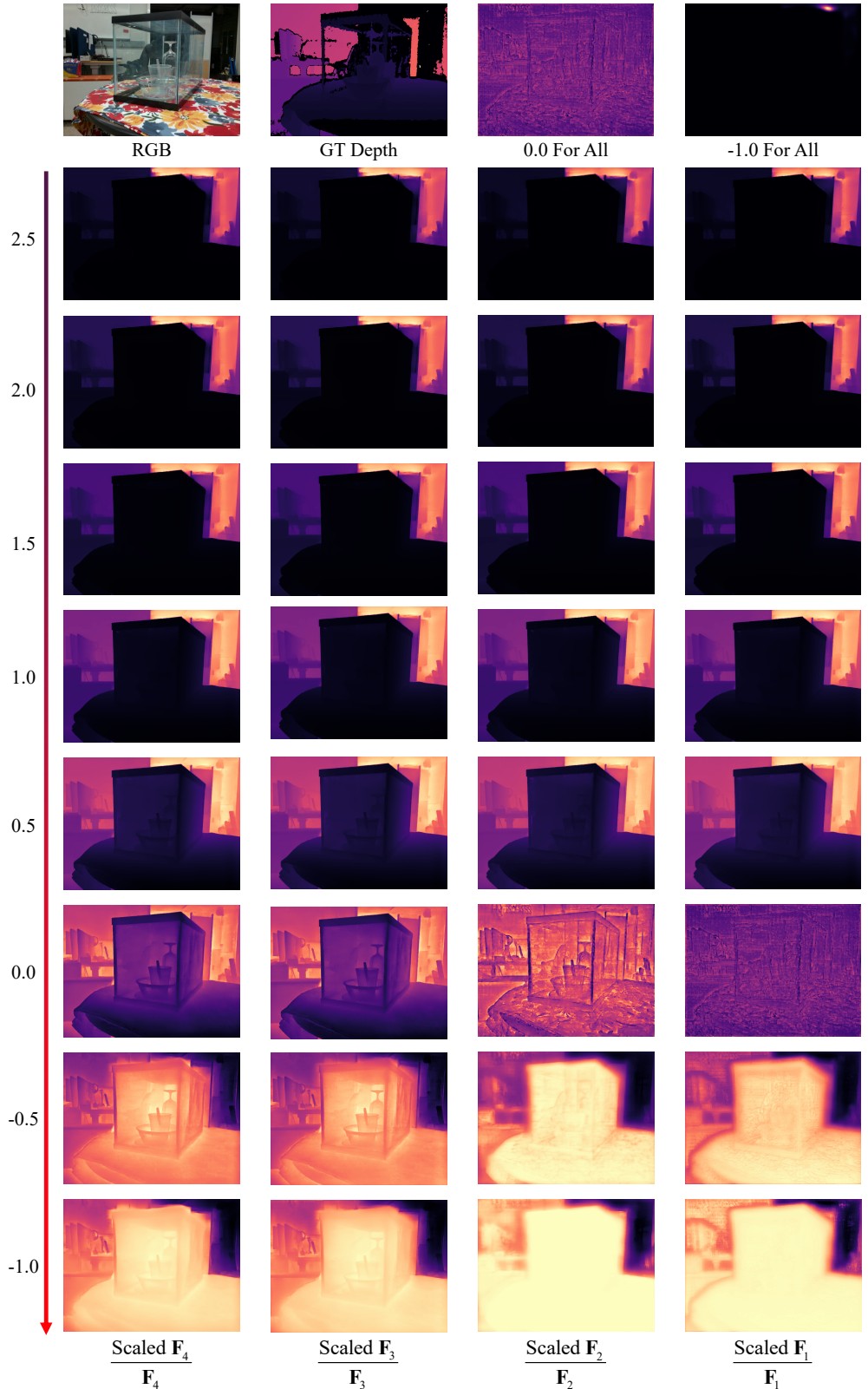

Figure 11: Comparison of different fusion weights on ClearPose dataset. Each column is independent and varies the corresponding weight from 1.5 to -2.0.

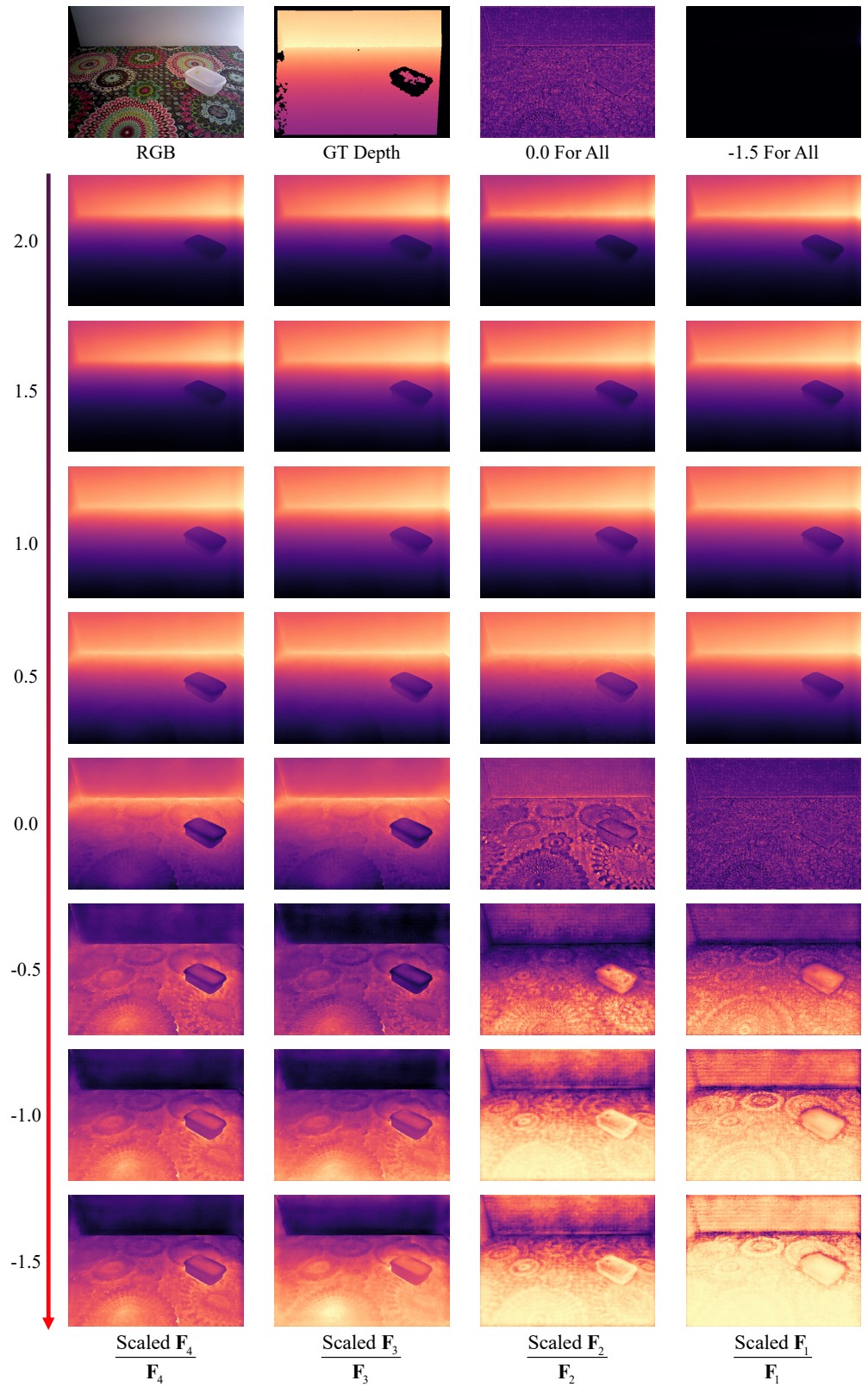

Figure 12: Comparison of different fusion weights on the Tyo-L dataset. Each column is independent and varies the corresponding weight from 1.0 to -2.5.

