# OpenReview forum: "SingRef6D: Monocular Novel Object Pose Estimation with a Single RGB Reference"
_NeurIPS.cc/2025/Conference — NeurIPS 2025 poster_

### Official Review · Reviewer_QjZ6 · 2025-06-24

**Clarity:** 4
**Significance:** 3
**Originality:** 2
**Rating:** 5
**Confidence:** 4

**Summary:**

The paper proposes SingRef6D, a novel object pose estimation method based on monocular image and corresponding monocular depth estimation. The method is composed of two parts. Depth estimation part Pose estimation part.

Depth estimation is based on Depth Anything V2, but includes an extra part that is meant for improving depth quality as well as adapting the scale of the depth to the given training scenes.

The pose estimation part is composed of two parts, LoFTR-based matching and PointDSC-based refinement. The proposed method modifies LoFTR to take depth, which improves the quality of matching further, especially on the photometrically challenging material object, like glasses.

The paper shows an experiment on depth estimation to prove the effectiveness of the fine-tuning method proposed in the paper. also pose estimation, mainly based on a matching pipeline, to prove the performance of the proposed matching method with depth.

**Questions:**

1. Misleading term with "Metric Depth Prediction"

- It feels like fine-tuning is meant to overfit the depth prediction on each dataset's distance instead of actually managing to predict the correct metric depth for different conditions. For example, the scaler is fine-tuned on the ClearPose dataset again before running the validation, which means that depth prediction is only a metric when it's fine-tuned to the specific scenario. It's rather "overfitting" on the specific scenario than predicting the correct metric depth.
- An interesting experiment will be using C.2's depth prior to make the pipeline compatible with different scenes without finetuning to show the robustness of the proposed network on different scenarios.

2. Missing quantitative evaluation

- I am curious why some of the experiments shown in Fig.4 are missing in Tab.2. For example, SPIDepth, ScaleDepth, Metric3D results are missing in Tab.2. Is it because they are significantly worse? It would be great to show their number regardless.

3. The experiment on the reflective objects is missing, although it is mentioned in the abstract and intros.

- There is one depth dataset [1] that is based on the same tabletop scenario with household objects, including reflective items, as well as high-quality depth ground truth, so that the author can fine-tune the proposed method. Also, a 6D pose dataset [2] that contains photometrically challenging objects, both transparent and reflective. It will be great to run an experiment on the reflective items by first fine-tuning the method on the depth dataset [1] and running pose estimation on the pose dataset [2].

4. The experiment is weak on the 6D pose estimation part.

- In lines 47-48, it is mentioned that the proposed method can perform competitively compared to generative or reconstruction-based methods. However, the experiment only covers matching-based methods. It would be great to include a comparison of such methods in the experiment section (e.g,  Any-6D, Zero123-6D, etc) by selecting two frames (i.e., one frame for generation, the other frame for getting pose). It's possible that the proposed method underperforms compared to such methods, but it's fair, as the proposed method is light. This experiment can give a better idea of the trade-off between complexity and performance.  Also, the experiment section lacks experiments on methods that are specifically targeted and trained on pose estimation. Including such experiments will strengthen the experiment section of the paper.


[1] On the Importance of Accurate Geometry Data for Dense 3D Vision Tasks, Jung et.al.

[2] Housecat6D: a large-scale multi-modal category level 6D object perception dataset with household objects in realistic scenarios, Jung et. al.

**Ethical Concerns:**

["NO or VERY MINOR ethics concerns only"]

**Final Justification:**

The authors ran extra experiments requested by reviewers with reasonable results, and also address the questions fairly well.

**Limitations:**

Yes

**Quality:**

3

**Strengths And Weaknesses:**

Strength :
1. The paper is easy to read, the figures are intuitive, and shows good extra details in the supplementary material.
2. Both qualitative and quantitative experiments show the superior performance of the proposed method
3. The proposed method specifically focuses on the photometrically challenging material, such as transparent and reflective.

Weakness (Suggestions on each points are written in the Questions section):
1. Misleading term with "Metric Depth Prediction"
2. Missing quantitative evaluation on few depth methods shown in qualitative evaluation
3. The experiment on the reflective objects is missing, although it is mentioned in the abstract and intros.
4. The experiment is weak on the 6D pose estimation part.
5. Contribution is slightly weak - The paper's main contribution is rather upgrading and fine-tuning depth prediction, adding depth in LoFTR matching. The rest of the pipeline is rather combination of existing works.

---

> ### Author Rebuttal · Authors · 2025-07-31
>
> Thank you for the positive feedback. We’re glad our clear presentation, intuitive figures, and detailed supplementary material were well received. We appreciate the recognition of our method’s strong performance, especially on challenging materials like transparent and reflective surfaces often overlooked in prior work. Below, we summarize and respond to each question, hoping to address your concerns.
>
> > **Q1:** The term "Metric Depth Prediction" seems misleading — the model is fine-tuned to each dataset's scale, effectively overfitting to specific scenarios rather than truly predicting metric depth across varied conditions. An interesting experiment will be using C.2's depth prior to make the pipeline compatible with different scenes without fine-tuning to show the robustness of the proposed network on different scenarios.
>
> **A1:** We appreciate your insightful comment. We fine-tuned our model on the ClearPose training set, but no calibration or optimization was performed on the test set scenes. This ensures that the backgrounds and environments in the test set remain novel and unseen during training. Since the model directly outputs metric depth, we refer to it as metric depth prediction. We appreciate your point; it might be more accurate to describe our approach as metric-aware adaptation, and we will consider using such terminology in revisions.
>
> To investigate overfitting, as shown in Table 2 of the main paper, directly fine-tuning a vanilla DPAv2 on REAL275 significantly degrades performance ($\delta_{1.05}$ drops to 29.87), whereas our method achieves a higher result of 44.28 (even with less data), demonstrating better robustness. This highlights the advantage of our token scaler in preserving generalizable representations by adaptively integrating hierarchical features rather than overfitting to dataset-specific distributions. Moreover, since our approach reuses the pre-trained model via a zero-initialized projection and layer-wise modulation, the original knowledge is well retained, mitigating significant alterations for the weights. Given the strong spatial understanding capabilities of transformer-based models like VGGT, we are also interested in extending our method to such architectures, as our token scaler is readily applicable to other hierarchical vision transformers.
>
> We agree with you that it can be used for tuning-free inference. Therefore, we conducted experiments using the depth prior defined in Section C.2 to scale DPAv2's predicted depth without dataset-specific fine-tuning.
> |Depth|AR↑|VSD↑|MSSD↑|MSPD↑|ADD↑|
> |-|-|-|-|-|-|
> |Manual|32.4|10.1|38.8|48.2|33.4|
> |Raw|1.4|0.2|1.2|2.8|2.0|
> |DPAv2|9.1|1.6|11.1|14.4|3.9|
> |DPAv2(Raw as Prior)|11.8|2.4|10.9|22.1|6.4|
> |Ours|**19.4**|**5.1**|**19.7**|**33.5**|**10.0**|
>
>
> As shown in the table, using raw depth priors improves performance over original DPAv2 predictions by enforcing global scale consistency. However, it still lags behind our full pipeline, as raw depth often fails to capture accurate object geometry and spatial layout, particularly in challenging cases like transparent objects in ClearPose. Even with depth alignment, DPAv2 struggles with surface reconstruction errors. That said, it is highly affected by the sensor and pretraining.
>
> In contrast, our method enhances geometric understanding through losses such as normal consistency and edge emphasis. Our token scaler adapts features based on scene structure, improving robustness across diverse object categories. These advantages with a depth-aware matching yield stronger performance, critical for accurate 6D pose estimation. To the best of our knowledge, we are the first that propose using depth prior to scale the predicted depth for pose estimation within our settings. During the practical implementation, it is beneficial to first try depth prior. In the revised version, we will include additional experiments using various depth models and scaling strategies to validate the generalization of our method. We also aim to explore a tuning-free approach adaptable to diverse scenarios.
>
>
> > **Q2:**   I am curious why some of the experiments shown in Fig.4 are missing in Tab.2. For example, SPIDepth, ScaleDepth, and Metric3D results are missing in Tab.2. Is it because they are significantly worse?
>
>
> **A2:** Thank you for your valuable suggestion. In Table 2, we have fine-tuned these methods for fair comparison during the experiment history. For example, on the REAL275 dataset, the fine-tuned SPIDepth achieves a $\delta_{1.05}$ of 11.09, likely due to its reliance on large-scale outdoor training data. The core idea of ScaleDepth is to decompose metric depth estimation into two stages: scale prediction and relative depth estimation. However, under our current data budget and scene types, this strategy is more sensitive to the convergence of scale distribution, which leads to weaker performance on $\delta_{1.05}$ of only 11.48, which is lower than other methods. Metric3D uses a CSTM module to address scale ambiguity caused by varying camera intrinsics. However, depth estimation for transparent objects is challenging due to background interference and refraction, leading to additional scale ambiguity. As shown in Figure 4, Metric3D performs poorly on the ClearPose dataset. Thus, Metric3D is better suited for depth estimation of regular objects, while transparent and highly reflective objects (which are considered in our task) are beyond its scope. Moreover, its heavier architecture makes the fine-tuning challenging, which only achieves 10.6 in $\delta_{1.05}$ on REAL275. Therefore, we initially omitted these results. However, based on your comment, we are willing to include them in the revision.
>
> > **Q3:**   It will be great to run an experiment on the reflective items in dataset [1] and dataset [2].
>
> **A3:** We sincerely thank the reviewer for the insightful suggestion. In the supplementary material, we include videos demonstrating depth prediction results on reflective objects (e.g., a metal spoon), which partially showcase our method's robustness to challenging surfaces. We have carefully reviewed the two referenced works and their corresponding datasets. Both the HAMMER dataset [1] and HouseCat6D [2] are high-quality resources. We agree with you that incorporating them would further validate our approach. To provide a quick analysis, we fine-tuned our depth model on a subset of the training data and evaluated 6D pose estimation on the validation subset of HouseCat6D. We manually selected three reflective objects (teapoint, knife, and cup) as targets and generated 100 image pairs for each.
> |Matcher|Depth|AR↑|VSD↑|MSSD↑|MSPD↑|ADD↑|
> |-|-|-|-|-|-|-|
> ||Oracle|57.4|41.0|62.6|68.7|68.8|
> |Oryon|DPAv2|26.6|9.1|33.6|37.3|39.5|
> ||Ours|38.1|20.8|44.4|49.3|49.0|
> ||Oracle|63.2|48.2|73.0|78.6|76.2|
> |Ours|DPAv2|30.1|12.0|41.9|46.3|43.8|
> ||Ours|**44.5**|**26.1**|**51.8**|**55.5**|**55.2**|
> |vs.Oryon|Δ|+6.4|+5.3|+7.4|+6.2|+6.2|
>
> Our method outperforms the baseline and yields higher depth quality than fine-tuned DPAv2. Baselines like Oryon rely solely on RGB, which the HouseCat6D challenges through images with reflective objects, causing varying appearances. This makes matching solely based on RGB unreliable.
>
> In contrast, our coarse-to-fine matching leverages depth cues for spatial consistency, enabling the model to access geometric relationships even when RGB is unreliable, thus boosting accuracy. High-resolution feature maps further refine correspondences, reducing errors in tough cases. These results showcase both the architectural strength and real-world applicability of our approach under varying RGB conditions.
>
> We plan to further validate our approach by systematically fine-tuning on the full HAMMER dataset and evaluating pose estimation across a broader range of categories in HouseCat6D. All scripts and splits will be released to reproduce the evaluation.
>
>
> > **Q4:**   It would be great to include a comparison of such methods in the experiment section (e.g., Any-6D). Also, the experiment section lacks experiments on methods that are specifically trained on pose estimation. It's possible that the proposed method underperforms compared to such methods, but it's fair, as the proposed method is light.
>
> **A4:**  We appreciate your valuable advice. We admit works like Any-6D are high-quality contributions. In response, we have conducted additional comparisons on Tyo-L, and the results are in the table below.
> |Matcher|Depth|AR↑|VSD↑|MSSD↑|MSPD↑|ADD↑|
> |-|-|-|-|-|-|-|
> ||Oracle|43.3|15.8|55.8|58.4|32.2|
> |Any-6D|DPAv2|5.2|1.3|4.3|11.0|2.2|
> ||Ours|31.6|13.5|33.5|47.8|13.2|
> ||Oracle|42.0|34.2|44.1|47.6|35.0|
> |Ours|DPAv2|5.8|2.8|3.5|11.1|2.9|
> ||Ours|31.7|13.9|33.8|47.3|13.1|
>
> While Any-6D achieves slightly better results with ground truth depth, it benefits from its strong image-to-3D reconstruction model; it heavily relies on depth quality—its performance drops with DPAv2-predicted depth. Due to its global optimization (e.g., point cloud registration) improves MSSD/MSPD, but might accumulate local errors due to predicted depth, lowering VSD recall. In contrast, our method conducts a depth-aware image-based alignment, avoiding too much dependency on 3D registration with all points, achieving higher VSD recall.
>
> We also reproduced results of methods like FoundationPose (77.9 AR on LM-O) and SAM6D (68.0 AR), which outperform ours but under distinct settings. FoundationPose needs a neural field over a full 3D scene and trains for a week on 4 V100 GPUs, and SAM6D uses dense multi-view templates from CAD models with training on 2 GPUs for 6 days. In contrast, our lightweight method fine-tunes on a single GPU using only one RGB reference image, possibly from a different scene, without implicit representation or CAD templates, and it greatly reduces computational and data requirements. We appreciate the advice and will include this discussion in the revised manuscript to clarify our method’s positioning and strengths.

---

> > ### Comment · Reviewer_QjZ6 · 2025-08-08
> >
> > Thanks to the authors for providing the extra experiments that I requested. Overall, I see the results of the additional experiments and resolved my questions, and also proved that some claims that was not addressed in the paper (e.g. for reflective objects). I already wrote my final ratings with short comment, but I asusme that was not enough, so I leave my comment here.

---

> > > ### Author Response · Authors · 2025-08-09
> > >
> > > Thank you for your follow-up comment and for acknowledging the additional experiments in our rebuttal. We are glad that these results have addressed your questions. We will incorporate the results from the rebuttal into the revised version.

---

### Official Review · Reviewer_dpvV · 2025-06-29

**Clarity:** 4
**Significance:** 3
**Originality:** 3
**Rating:** 5
**Confidence:** 4

**Summary:**

This paper presents SingRef6D, a method for novel object 6D pose estimation that only requires a single RGB reference image. The authors propose a lightweight pipeline that combines a fine-tuned Depth-Anything v2 model with a novel token-scaler mechanism for depth prediction and integrates depth-aware matching into LoFTR to improve correspondence estimation. The proposed method does not require CAD models, novel-view synthesis, or video-based supervision, making it efficient and practical. Experimental results on REAL275, ClearPose, and Toyota-Light demonstrate strong performance and improvements over existing works.

**Questions:**

- How does the proposed method perform when there is a large viewpoint gap between the reference and query images—specifically, in cases where the two views have minimal overlap?
- How robust is the proposed approach to heavy occlusion, such as in Occlusion-Linemod dataset?

**Ethical Concerns:**

["NO or VERY MINOR ethics concerns only"]

**Final Justification:**

I have read through all the reviews and the rebuttal. Overall, I believe the authors have responded well to the points raised, particularly with the addition of new supporting experiments. Therefore, I keep my score and recommend accepting this paper.

**Limitations:**

I do not see any major limitation in this work. While the authors mention the use of object masks for target localization, I do not consider this to be a significant limitation. In practice, object masks can be easily obtained using existing tools such as SAM or SAM2. Moreover, since there are typically multiple objects present in a scene, it is entirely reasonable to specify which object the model should focus on in the reference image.

**Paper Formatting Concerns:**

The paper does not have any issues related to formatting.

**Quality:**

3

**Strengths And Weaknesses:**

Strengths:
- The problem addressed in this paper is well motivated. Estimating 6D pose from a single RGB image is very challenging but highly practical for real-world applications.
- I like the idea of using depth models for this pose estimation task, as recovering object scale from only a single RGB reference is highly ambiguous. The proposed token-scaler mechanism further improves the robustness and accuracy of metric depth prediction on challenging surfaces, such as transparent or reflective materials. The reported gains in depth metrics across multiple datasets are significant.
* The model is computationally efficient compared to recent baselines like Oryon and RoMA, requiring fewer parameters and less memory while achieving better or comparable accuracy.
* Extensive evaluations and ablation studies are provided, which clearly demonstrate the effectiveness of each proposed component, including the depth losses and feature fusion strategies.

Weaknesses:
* One major concern in estimating 6D pose from a single RGB reference is when there is a large viewpoint gap between the reference and the query image—i.e., when the two views share only a small overlap. Although the authors show promising qualitative examples (e.g., the first row of Figure 5), I would expect to see a more thorough quantitative evaluation that directly studies this challenge. If the proposed method is robust in small-overlap conditions, that would be a very strong result to highlight in the paper.
* While the paper includes difficult examples such as transparent objects, most of the presented samples involve objects that appear relatively clear, with minimal occlusion. It is unclear how well the proposed method handles more severe occlusions, such as those found in the Occlusion-LM dataset. The authors could consider using the single-image setup introduced in GigaPose [39] for a more comprehensive comparison.

---

> ### Author Rebuttal · Authors · 2025-07-31
>
> We thank the reviewer for the encouraging feedback. We're encouraged that you found our problem motivation clear and recognized the practical value of our SingRef6D. We appreciate your recognition of our use of depth models to resolve scale ambiguity and the token-scaler mechanism for improving depth prediction on challenging surfaces. Your acknowledgement of our significant performance gains, computational efficiency over strong baselines like Oryon and RoMA, and the thoroughness of our evaluations is truly appreciated. We summarize and respond to each question below, and hope our answers address your concerns.
>
> > **W1/Q1:** How does the proposed method perform when there is a large viewpoint gap between the reference and query images, specifically, in cases where the two views have minimal overlap?
>
> **A1:** We appreciate your insightful suggestion. We agree with you that quantifying such a relationship is beneficial for our work. To this end, we conduct an experiment that manually rotates the scene with a specific angle $\alpha \in [15^\circ,150^\circ]$  along the $X$,$Y$, and $Z$ axes, respectively. Specifically, we use accessible 3D models (e.g., from BOP datasets) to construct a scene with BlenderProc and rotate the camera by $\alpha$ (The reference scene and the query differ in background and contain different objects, except for the target object). We define a right-handed coordinate system, where the $Z$-axis points vertically upward, perpendicular to the object surface, and the $X$ and $Y$ axes lie in the horizontal plane. The camera is positioned at (1, 1, 1), measured in meters, looking down at the scene, with all objects placed on the $X$–$Y$ plane. Since most existing benchmarks adopt a top-down perspective, we follow the same setup. To evaluate the effect of viewpoint variation, we report results under different camera rotations along the $X$ and $Z$ axes. Rotation along the $Y$ axis is omitted, as we observed a strong correlation between rotations around the $X$ and $Y$ axes during the experiments, with a Spearman coefficient exceeding 0.86. This correlation is likely due to the symmetry of the setup. Therefore, we report only the $X$-axis rotation in the table below to represent horizontal-axis variations. In future revisions, we plan to include finer-grained angular distinctions and explicit values for $Y$-axis rotations on more scenes.
>
> |Axis\α|15°||30°||45°||60°||75°||90°||120°||150°||
> |-|-|-|-|-|-|-|-|-|-|-|-|-|-|-|-|-|
> ||AR|ADD|AR|ADD|AR|ADD|AR|ADD|AR|ADD|AR|ADD|AR|ADD|AR|ADD|
> |X|69.84|65.70|61.17|60.42|50.52|50.38|42.36|42.91|33.71|33.64|30.12|30.47|26.83|22.36|20.54|19.90|
> |Z|74.26|72.19|71.50|69.83|68.13|67.21|65.80|64.02|62.42|60.57|58.31|56.48|39.76|36.73|31.95|28.54|
>
> From the table, we observe that pose estimation performance degrades with increasing viewpoint differences between the reference and query images, especially when the rotation is along the X-axis. This is because X-axis rotations significantly alter perspective and occlusion patterns, making it more challenging for the model to match spatial features. In contrast, Z-axis rotations mainly induce in-plane transformations with relatively mild geometric distortion, leading to a more gradual performance drop. However, when the rotation exceeds 90°, even Z-axis changes cause noticeable degradation. This is likely due to decreased visual discriminability — for example, the front of a mug may have a distinct texture or logo, while the back is plain, reducing the object's appearance discriminability and making it harder for the model to rely on appearance and forcing it to depend solely on geometry and spatial consistency.
>
> Despite this challenge, our method exhibits a smaller performance drop (65%) under such extreme conditions, while baselines like Oryon and SIFT fail almost entirely, with performance degradation exceeding 90\%. This robustness stems from our model’s ability to effectively exploit spatial consistency and adapt spatial representations through the proposed token scaler, thus boosting geometric understanding in the coarse-to-fine matching process. This mechanism helps to improve matching accuracy when RGB cues are unreliable. Furthermore, our loss functions impose strong geometric constraints such as normal consistency and edge emphasis, enabling the model to maintain a reasonable estimation accuracy even under large viewpoint changes. These results and analyses demonstrate the robustness of our model and highlight its potential for real-world applications. We also believe this direction presents a promising avenue for future research.
>
> > **W2/Q2:**  It is unclear how well the proposed method handles more severe occlusions, such as those found in the Occlusion-LM dataset. The authors could consider using the single-image setup introduced in GigaPose for a more comprehensive comparison. How robust is the proposed approach to heavy occlusion, such as in the Occlusion-Linemod dataset?
>
>
> **A2:** We appreciate your insightful advice. The authors agree with you that considering occlusion is important. We conducted two additional experiments on LM-O and YCB-V, which are representative benchmarks featuring occlusions and weakly textured objects, such as monochromatic, single-material objects. We evaluate pose estimation on three different target objects, generating 120 image pairs for each. The table below illustrates the experiment results.
> |Dataset|Matcher|Depth|AR↑|VSD↑|MSSD↑|MSPD↑|ADD↑|
> |-|-|-|-|-|-|-|-|
> ||Oryon|Oracle|6.1|3.1|6.9|8.3|5.5|
> |||Ours|3.8|1.9|4.2|5.4|2.7|
> ||Ours|Oracle|9.9|4.3|11.5|13.9|8.7|
> |LM-O||Ours|**5.3**|**2.6**|**5.7**|**7.3**|**3.8**|
> ||vs.Oryon|Δ|+2.7|+0.95|+3.1|+3.8|+2.2|
> ||Oryon|Oracle|8.6|4.7|9.3|11.8|7.5|
> |||Ours|5.8|2.4|6.6|8.4|5.2|
> ||Ours|Oracle|11.4|5.2|12.3|15.7|8.9|
> |YCB-V||Ours|**7.9**|**3.7**|**8.5**|**11.6**|**6.4**|
> ||vs.Oryon|Δ|+2.45|+0.90|+2.45|+3.55|+1.30|
>
> From experimental outcomes, heavy occlusion still results in a notable performance decline. Nonetheless, our method has all-around superiority compared to the baseline. This is due to our framework introducing a human vision-inspired mechanism that adaptively adjusts features across different layers, guided by both appearance and spatial cues. Unlike conventional methods such as Oryon or SIFT that rely heavily on RGB similarity, our matching process integrates spatial information to enhance alignment fidelity. Moreover, it considers spatial consistency in both coarse-matching and fine-grained matching (in Figure 1). These designs lead to significantly improved robustness, especially under challenging conditions where appearance-based cues alone are insufficient (e.g., occlusion), which validates that our performance is not accidental but benefits from the designed mechanism.
>
> Methods like GigaPose and Any-6D rely on a single-image 3D reconstruction model, which makes them highly sensitive to object texture and the quality of the reconstruction model. In contrast, our approach is lightweight and based on cross-image matching, making it more robust to texture variations and easily integrable with other multi-view pose estimation pipelines. We are open to including additional comparisons with such methods in the revised version and plan to explore their integration in future work.

---

> > ### Comment · Reviewer_dpvV · 2025-08-05
> >
> > Thank you to the authors for thoroughly addressing the concerns raised by myself and the other reviewers. I have read through all the reviews and the rebuttal. Overall, I believe the authors have responded well to the points raised, particularly with the addition of new supporting experiments.

---

> > > ### Author Response · Authors · 2025-08-05
> > >
> > > Thank you for your time and thoughtful feedback during the review process. We’re glad that our rebuttal addressed your concerns and are especially grateful for your recognition of our work. Your suggestions are highly valuable, and we are committed to incorporating them to enhance the final version of our work. Thank you again for your constructive comments and support.

---

### Official Review · Reviewer_Xucn · 2025-07-02

**Clarity:** 3
**Significance:** 2
**Originality:** 2
**Rating:** 4
**Confidence:** 4

**Summary:**

This paper presents a complete pipeline for novel object pose estimation from a single RGB image. The core methodology consists of three main stages: （1）Scaled Depth Estimation: The authors first propose a fine-tuning strategy based on a token-scaler architecture. This method adapts a monocular depth estimation model to recover a scene depth map with a correct metric scale from a single RGB input. （2）Depth-guided Feature Matching: Subsequently, the paper employs a training-free feature matching technique (LoFTR) to establish correspondences for the object instance between a reference and a query image. A key point here is the use of the previously estimated depth map to guide this matching process, which aims to improve the accuracy of the resulting keypoint matches. （3）Pose Recovery: Finally, using these depth-guided matches, the relative pose between the reference and query views is solved for, which in turn yields the final pose of the object in the query image. The proposed method was evaluated on three benchmark datasets. Through comparisons with various combinations of different depth estimation models and feature matching techniques, the experimental results demonstrate that the proposed integrated approach achieves promising results for the task of novel object pose estimation.

**Questions:**

1. How about the performance of the proposed metric depth estimation model on standard depth benchmarks？Within the presented object pose estimation experiments, what is the degree of similarity or difference between the scenes used for fine-tuning the depth model and the scenes used for testing?

2. Could you please provide a quantitative comparison of the proposed method against recent works[A,B] on a standard instance-level benchmark (e.g., LINEMOD)?

3. Could you please provide an analysis of your method's robustness to the choice of the reference view? Specifically, in your current quantitative results, what is the typical pose difference (e.g., in terms of rotation) between the reference and query images? How does performance degrade as this difference increases?

**Ethical Concerns:**

["NO or VERY MINOR ethics concerns only"]

**Final Justification:**

The rebuttal and discussion addressed most of my concerns. Therefore, I will raise my score to Borderline Accept. At the same time, I maintain my position that the paper is only suitable for acceptance as a contribution to object pose estimation if, and only if, the authors incorporate the following crucial experiments from their rebuttal into the main manuscript. Specifically, the paper must include: （1）A cross-domain generalization experiment for pose estimation.
（2）A direct comparison with recent relevant methods on instance-level pose estimation benchmarks.
（3）A comparison with point cloud-based registration methods.
（4）A study on the robustness of pose accuracy to the selection of the reference view. My recommendation is contingent upon these revisions, assuming such changes are permissible within the NeurIPS camera-ready policy.

**Limitations:**

Yes

**Quality:**

2

**Strengths And Weaknesses:**

Pros：

+ The paper outlines a complete framework for novel object pose estimation using a single RGB image. The methodology is presented as a sequence of three main modules: monocular metric depth estimation, depth-guided feature matching, and final pose recovery. This modular structure clearly defines the paper's technical approach.

+ The authors introduce a fine-tuning strategy for a monocular depth model aimed at recovering the depth metric. They conduct experiments comparing this method against other approaches, and the paper reports that their method leads to improved depth accuracy and reduced geometric distortion.

+ The proposed pipeline is evaluated on three public datasets. The paper presents quantitative results for the pose estimation task, including experiments on the ClearPose dataset, which contains challenging transparent objects.

Cons:

- The core technical contribution appears to be the fine-tuning strategy for metric depth recovery. The proposed depth estimation method is a solution to a well-established problem (metric depth from a single image). As such, it should be also evaluated on standard depth estimation benchmarks. The current evaluation, which is confined to the three object pose datasets, does not provide a clear picture of how the method performs in a general context or how it compares to state-of-the-art depth estimation techniques. Besides, the paper seems to imply that the depth model is fine-tuned separately for each of the three datasets. This raises significant concerns about the model's generalization capabilities. It is unclear how different the fine-tuning scenes are from the testing scenes. A crucial unanswered question is: how well would a model fine-tuned on a general indoor dataset (e.g., ScanNet) perform when applied to a completely different scene with a similar scale? Without this analysis, the practicality of the proposed depth recovery method remains questionable.

- From a pose estimation perspective, the paper's contribution is limited. The overall pipeline that uses feature matching followed by a PnP/relative pose solver is a standard and well-established practice. The main novelty claimed for the pose estimation part is the integration of depth to guide the LoFTR feature matcher. While this addition appears effective, the idea of using geometric priors (like depth) to refine feature matching is not new. The specific implementation is straightforward and can be seen as an incremental improvement rather than a novel conceptual contribution. Given that the method explicitly recovers depth for both the reference and query images, how does the proposed 2D matching with depth guidance compare to methods that perform keypoint matching directly on the generated point clouds, or use techniques like ICP for pose refinement? This comparison is currently missing.

- The paper's experimental setup is more aligned with instance-level relative pose estimation. Therefore, the evaluation should also be extended to standard instance-level benchmarks like LINEMOD and LINEMOD-Occlusion. In addition,  the comparisons should also include recent and highly relevant instance-level methods [A,B].

[A] "3D-Aware Hypothesis & Verification for Generalizable Relative Object Pose Estimation", ICLR 2024.

[B] "DVMNet: Computing Relative Pose for Unseen Objects Beyond Hypotheses", CVPR 2024.

- As a reference-based method, the choice of the reference image is critical to performance, yet this aspect is not analyzed. The paper does not provide any analysis on how the performance varies with the choice of the reference view. What is the typical viewpoint difference (e.g., in terms of rotation and translation) between the reference and query images in the reported experiments? The robustness of the feature-matching-based approach is highly dependent on the visual overlap between the two views. It is expected that the performance will degrade significantly as the viewpoint difference increases. A thorough ablation study is needed to quantify this effect and understand the method's limitations, which is a standard practice for evaluating reference-based methods.

---

> ### Author Rebuttal · Authors · 2025-07-31
>
> Thank you for the thoughtful feedback. We’re glad our modular framework for novel object pose estimation is clear and technically sound. We appreciate your recognition of our improvements in depth accuracy, geometric consistency, and thorough evaluation across diverse datasets, including challenging transparent objects in ClearPose. We summarize and address each of the questions and hope they can address your concerns.
>
> > **W1:** How well would a model fine-tuned on a general indoor dataset perform when applied to a completely different scene with a similar scale?
>
> **A1:** Thank you for the comment. Although we aim to enable quick adaptation through efficient fine-tuning on relevant scenes, we agree that cross-domain generalization is important. We fine-tuned the depth model on REAL275 and evaluated it on ClearPose and Tyo-L without further adaptation. These datasets differ in scale, background, camera settings, and objects, offering a rigorous test of generalization (can be treated as distinct benchmarks).
> |Setting|Method|$\delta_{1.05}$↑|$\delta_{1.10}$↑|$\delta_{1.25}$↑|RMSE↓|log10↓|Abs.Rel↓|Sq.Rel↓|MAE↓|
> |-|-|-|-|-|-|-|-|-|-|
> ||Unidepth|8.35|22.61|42.01|0.983|1.419|1.855|0.702|0.500|
> |REAL275→Tyo-L|DPAv2|10.55|28.93|50.23|0.519|0.764|0.399|0.177|0.326|
> ||Ours|**28.13**|**34.91**|**45.72**|**0.377**|**0.471**|**0.223**|**0.119**|**0.218**|
> ||Unidepth|8.74|14.02|25.91|0.364|0.268|0.294|0.312|0.273|
> |REAL275→ClearPose|DPAv2|20.85|44.48|62.36|0.258|0.232|0.207|0.280|0.181|
> ||Ours|**42.19**|**66.25**|**79.70**|**0.182**|**0.106**|**0.131**|**0.228**|**0.109**|
>
> The experimental results show the expected drop in cross-domain performance. However, our method consistently outperforms baselines under identical conditions. This benefits from our Token Scaler’s data-agnostic feature modulation, which improves generalization ability across diverse inputs.
>
> Our loss functions enhance geometric consistency by leveraging generalizable cues like edges and surface normals, avoiding overfitting to specific datasets. These improvements stem from the model’s intrinsic design rather than domain-specific tuning. Besides, the proposed modules are easily transferable to other visual tasks (e.g., depth-based navigation), demonstrating broad applicability. We will include these cross-domain results in the revised version to support our claim.
>
>
> > **W2:**  The specific implementation is straightforward and can be seen as an incremental improvement. How does the proposed matching with depth guidance compare to methods that perform keypoint matching directly on the generated point clouds, like ICP?
>
> **A2:** Thank you for the helpful advice. We admit ICP is a baseline worthy of comparison. Our setting focuses on pose estimation from a single reference view, which often yields incomplete point clouds, so we initially considered ICP out of scope. Nevertheless, we conducted ICP experiments using point clouds from both reference and query views. The results are shown in the table below.
> |Matcher|Depth|AR↑|VSD↑|MSSD↑|MSPD↑|ADD↑|
> |-|-|-|-|-|-|-|
> ||Manual|19.8|5.5|23.2|32.9|20.6|
> |ICP|Raw|0.3|0.04|0.25|0.61|0.9|
> ||Ours|8.1|2.6|9.4|12.4|4.5|
> ||Manual|32.4|10.1|38.8|48.2|33.4|
> |Ours|Raw|1.4|0.2|1.2|2.8|2.0|
> ||Ours|**19.4**|**5.1**|**19.7**|**33.5**|**10.0**|
>
> ICP performs unsatisfactorily due to it is sensitive to initialization and noise, making correspondence unreliable. In contrast, our depth-aware matching operates in latent space, preserving pixel-level consistency and mitigating the pitfalls of noisy 3D reconstructions. Our method is enriched with spatial cues in a coarse-to-fine manner, further boosted following pose solving, performing better under transparent and low-texture conditions where ICP struggles.
>
> While we do not introduce a new matching conception—still following feature matching followed by pose estimation—our key contribution is embedding predicted dense depth into the latent space of the matching network. In the challenging setting of single RGB reference input, no CAD models, and pose-supervised training, this modular depth fusion consistently improves robustness to occlusion, weak textures, and transparency, all without retraining. We will include related discussions and results in the revised version.
>
>
> > **Q1:** How about the performance of the proposed metric depth estimation model on standard depth benchmarks? What is the degree of similarity or difference between the scenes used for fine-tuning the depth model and the scenes used for testing?
>
> **A3:** Thanks for your valuable comment. Our depth perception relies on the strong foundation model DPAv2. The proposed token scaler preserves its knowledge while dynamically modulating layer weights to adapt to different scenes. To avoid information leakage, we fine-tuned the token scaler (with DPAv2 pretrained on KITTI) on a downsampled NYU-D training set. On its test set, our method improves $\delta_{1.05}$ by **0.14** and reduces Abs.Rel by **0.09** compared to DPAv2 with a fine-tuned head. This shows our approach is effective for general depth prediction. We will include more results in the revised version.
>
>
> In our experiments, we manually split the data to ensure that test scenes remained entirely unseen during fine-tuning. To clarify this, we will add a data description section with visualizations in the revised version. To quantify the similarity between fine-tuning and test data, we compute the Structural Similarity Index Measure and report the mean and std for each dataset.
> |Dataset|SSIM|
> |-|-|
> |Tyo-L|0.247(±0.137)|
> |REAL275|0.167(±0.029)|
> |ClearPose|0.181(±0.041)|
>
> The SSIM scores are low, suggesting limited overlap and sufficient separation between scenes. In future work, we plan to explore more semantic-aware metrics, such as CLIP similarity or learned perceptual measures, to better assess cross-scene similarity.
>
> > **W3/Q2:** Could you provide a quantitative comparison of the proposed method against recent works[A, B] on an instance-level benchmark?
>
> **A4:**  Thanks for your insightful suggestion. The referenced works are indeed great works for instance-level relative pose estimation. They encode images into a trainable latent voxel space and achieve impressive results without relying on explicit depth inputs by learning from large-scale pose-annotated datasets. We will include a detailed discussion of these methods in the revised Related Work section. We compare with DVMNet and 3DVHA by performing iterative inference on image pairs. Results are summarized in the table below.
>
> |Method|AR|ADD|Angular Error|
> |-|-|-|-|
> |3DVHA|7.7|6.3|52.71|
> |DVMNet|8.6|7.4|51.66|
> |Ours|9.9|8.7|51.89|
>
> While our method shows slightly higher angular error than DVMNet, it achieves better AR and ADD, indicating more accurate translation estimation. This is because DVMNet and 3DVHA are supervised with full ground-truth poses for a latent voxel space, which might encounter translation shifts in an unseen environment.
>
> This shows a key difference: DVMNet is suited for stable scenes needing precise $SO(3)$ estimation, whereas our approach offers a more generalizable and lightweight solution for full-pose estimation, making it more practical for real-world applications. We will include these discussions and explore further experiments inspired by these notable works in the revised version.
>
> > **Q3:** Could you provide an analysis of your method's robustness to the choice of the reference view? In your current quantitative results, what is the typical pose difference (rotation) between the reference and query? How does performance degrade as this difference increases?
>
> **A5:** We appreciate your feedback. The table below reports the angle differences between reference-query pairs across datasets, showing reasonable variation rather than fixed or easily inferred values. This highlights the practical relevance of our setup and demonstrates our ability under various angular conditions.
>
> |Dataset|Range|Mean|Std.|
> |-|-|-|-|
> |REAL275|1.5-179.6|51.7|31.9|
> |ClearPose| 1.4-169.2|49.3|29.8|
> |Tyo-L|2.9-180.0|98.8|57.0|
>
> To assess the impact of viewpoint gaps between reference and query images, we conduct additional experiments in BlenderProc by constructing scenes using 3D models from BOP and applying rotations along the X, Y, and Z axes by angles $\alpha$. We report results for rotations along the X and Z axes. Y-axis rotations are omitted due to their strong correlation with X-axis rotations (Spearman>0.86), likely due to the symmetry of the setup. Future revisions will include finer angular splits and explicit Y-axis rotation results on more scenes.
> |Axis\α|15°||30°||45°||60°||75°||90°||120°||150°||
> |-|-|-|-|-|-|-|-|-|-|-|-|-|-|-|-|-|
> ||AR|ADD|AR|ADD|AR|ADD|AR|ADD|AR|ADD|AR|ADD|AR|ADD|AR|ADD|
> |X|69.84|65.70|61.17|60.42|50.52|50.38|42.36|42.91|33.71|33.64|30.12|30.47|26.83|22.36|20.54|19.90|
> |Z|74.26|72.19|71.50|69.83|68.13|67.21|65.80|64.02|62.42|60.57|58.31|56.48|39.76|36.73|31.95|28.54|
>
> The table shows that pose estimation performance declines with increasing viewpoint differences, especially along X-axis rotations, which introduce significant perspective and occlusion changes. Z-axis rotations mainly cause in-plane transformations with milder distortion, leading to a slower performance drop, though accuracy still degrades when rotations exceed 90°, likely due to reduced visual discriminability (e.g., textured front with logos vs. plain back of a cup).
>
> Despite these challenges, our method shows a 65% drop under extreme conditions, while baselines like Oryon and SIFT degrade by over 90%. This robustness comes from our model’s ability to exploit spatial consistency and adapt feature representations via the token scaler. Additionally, our loss functions enforce strong geometric constraints, helping maintain accuracy under large viewpoint changes. These findings underscore our method’s robustness and its potential for real-world applications, offering a promising direction for future research.

---

> > ### Comment · Reviewer_Xucn · 2025-08-03
> > **Official Comment by Reviewer Xucn**
> >
> > Thanks for the detailed answers, performed experiments, and additional results. Some of my initial concerns were addressed. However, based on the rebuttal, several important issues remained unclear to me:
> >
> > ### **Regarding the Response in A1**
> >
> > For the results presented in A1, I have the following two questions:
> >
> > 1.  The depth model, after being finetuned on the REAL275 indoor scenes, appears to suffer a **significant drop in depth accuracy** when tested on the Tyo-L and ClearPose indoor scenes. For instance:
> >     *   For **Tyo-L**, the $\delta_{1.05}$ metric drops from 80.09 to 28.13, and the RMSE increases from 0.112 to 0.377.
> >     *   For **ClearPose**, the $\delta_{1.05}$ metric drops from 54.30 to 42.19, and the RMSE increases from 0.103 to 0.182.
> >
> >     As this is a paper on object pose estimation, could the authors please clarify **how this degradation in depth accuracy impacts the final pose estimation results?**
> >
> > 2.  The results shown in A1 deepen my concerns about the validity of the **problem setting** adopted in this submission (monocular depth estimation + object pose estimation). The results seem to indicate that to apply the proposed method to a new target scene, one must first collect a dataset with ground-truth depth to finetune the depth estimation model before performing monocular RGB-based object pose estimation. The finetuned depth model does not appear to generalize well to other indoor scenes, even those with similar scales.
> >
> >     In this case, the proposed method does not seem to show a significant advantage over existing methods. Although the method does not require depth data at inference time, this comes at the **cost of needing to collect extra depth data for each new scene** to finetune the depth model. This cost might even be higher than that of conventional RGB-D or multi-view methods. Could the authors please elaborate on the advantages of the proposed framework from an object pose estimation perspective, specifically in terms of **scalability, generalization capability, or accuracy?**
> >
> > ### **Regarding the Response to A2**
> >
> > Apart from the conventional ICP, I am wondering about the comparative performance when compared with **learning-based point cloud keypoint matching or registration methods**.
> >
> > Moreover, what would the performance be if a learning-based point cloud registration method were used first to recover an initial pose, followed by ICP for refinement? Similar to the image feature matching method LoFTR that the submission used, there are many learning-based point cloud feature description/registration models that could be directly used. I believe this experiment is **necessary to validate the advantage** of the proposed depth-guided LoFTR matching scheme.
> >
> > ### **Regarding the Response to A4**
> >
> > Apologies if I missed this information, but it is unclear from the table **on which dataset the experimental comparison with 3DVHA and DVMNet was conducted**.
> >
> > Furthermore, what are the results when comparing against 3DVHA and DVMNet on **standard instance-level benchmark datasets, such as LineMOD and LineMOD-Occlusion?**
> >
> > ### **Regarding the Response to A5**
> >
> > Thanks for the clarification and experiments. Apart from the experiments on the additional BlenderProc scenes, a more straightforward analysis should be provided for the target **REAL275, ClearPose, and Tyo-L scenarios**.
> >
> > Could the authors please provide a quantitative breakdown of the pose estimation results on these three datasets, **binned by the query-reference viewpoint difference** (e.g., 0-15°, 15-30°, 30-45°, etc.)? Additionally, for measuring the viewpoint difference, please consider using the geodesic distance as presented in [Ref.1].
> >
> > [Ref.1] Metrics for 3D rotations: Comparison and analysis, Journal of Mathematical Imaging and Vision, 2009.

---

> ### Author Response · Authors · 2025-08-04
> **Response to Point 1 of New Question 1**
>
> > **New Q1 Point1:** The depth model, after being finetuned on the REAL275 indoor scenes, appears to suffer a significant drop in depth accuracy when tested on the Tyo-L and ClearPose indoor scenes. As this is a paper on object pose estimation, could the authors please clarify how this degradation in depth accuracy impacts the final pose estimation results?
>
> **Answer:** We are grateful for the opportunity to address your concerns and underscore the exceptional advantages of our proposed method.
>
> It is well-known that **cross-domain fine-tuning** (where the fine-tuning dataset differs significantly from the evaluation benchmark) can lead to performance degradation due to distribution shifts. This degradation often manifests in metrics such as $\delta_{1.05}$ and RMSE, which are highly sensitive to changes in global scale. However, in our pose estimation task, metrics like VSD, MSSD, and MSPD are more concerned with local geometric representations, edge stability, and relative depth consistency within the object. Therefore, a drop in global depth accuracy does not necessarily translate into a proportional decline in pose estimation performance.
>
> Specifically, our method **mitigates the effect of depth variations** on pose estimation through the proposed loss terms:
>
> - $\mathcal{L}_{scale}$ **preserves the relative scale** within the object to avoid internal structural distortion.
>
> - $\mathcal{L}_{edge}$ **stabilizes object boundaries** and foreground-background relationships, reducing jitter and ensuring accurate spatial representation.
>
> - $\mathcal{L}_{norm}$ **constrains surface normal** vectors to **avoid abrupt geometric changes** on the object’s surface.
>
> As shown in Table 6, these losses enhance the model's ability to preserve geometric consistency, leading to better depth estimation. As a result, our method achieves more stable performance in metrics such as VSD, MSSD, and MSPD compared to settings that use depth from vanilla fine-tuned DPAv2 and UniDepth, even under cross-domain fine-tuning.
>
> Apart from the geometric-aware losses, our method also maintains the foundational knowledge learned by the pre-trained model during cross-domain adaptation. This is achieved by freezing the parameters of LoFTR and DPAv2 and introducing a lightweight, zero-initialized Token Scaler, which preserves the pretrained spatial and geometric priors. As indicated in Table 7 of the paper, directly fine-tuning the depth head leads to inferior performance, while incorporating our Token Scaler improves stability with acceptable adaptation cost.
>
> We provide additional experimental results under the same setup and depth source: we fine-tune the depth model on REAL275 and evaluate pose estimation on Tyo-L. The results show that using the predicted depth from a vanilla fine-tuned DPAv2 results in a **32% increase in rotation error** and more than **60% decrease in Average Recall (AR)**. In contrast, our method achieves only an **11% increase in rotation error** and a **27% decrease in AR**. It demonstrates stronger robustness and generalization under cross-domain depth fine-tuning. Practically, we further enhance stability by applying a mask to restrict matching regions to the object’s region of interest (ROI), effectively ignoring noisy background regions. These results highlight the robustness of our approach in real-world scenarios. All experiments are conducted under the same setting with **A1**. We will release corresponding data and code in the future.

---

> ### Author Response · Authors · 2025-08-04
> **Response to Point 2 of New Question 1**
>
> > **New Q1 Point2:**. Could the authors please elaborate on the advantages of the proposed framework from an object pose estimation perspective, specifically in terms of scalability, generalization capability, or accuracy?
>
> Thanks for your feedback. We would like to emphasize that our setting only uses a single reference RGB image, without requiring additional pose annotations, CAD models, or multi-view supervision during training or inference. At inference time, our method operates without the need for a depth sensor. Instead of performing heavy scene-specific adaptation, we adopt a lightweight domain-level adaptation strategy. For example, a single fine-tuning on one scene with a similar scale is sufficient to generalize across other scenes within the same domain. Therefore, we argue that collecting a small dataset for efficient domain-level fine-tuning, which can subsequently generalize to different scenes within the same domain, is an economical and practical approach. Thus, enhancing the practicality and ease of deployment of our method in real-world scenarios.
>
> Compared to other baselines (such as DPAv2 with direct cross-domain fine-tuning), our approach maintains generalization capability under the same settings. Our method is able to effectively leverage a few data **(scalability)** while generalizing to previously unseen objects. More importantly, our approach has stable performance with single RGB references even for challenging cases, including transparent objects **(generalization)**, while demonstrating minimal relative performance degradation and better pose estimation results than baselines with cross-domain depth fine-tuning **(accuracy)**. Below, we elaborate on three advantages of the proposed method.
>
> First, owing to our proposed token scaler and geometric-aware losses, our method achieves better results than baselines in novel object pose estimation under the aforementioned setting. When using equivalent data volumes for fine-tuning and conducting pose estimation with our setting, the proposed method's performance surpasses other baselines as demonstrated in Table 3 and 4. This data efficiency stems from our token scaler's ability to preserve pre-trained knowledge while dynamically adjusting features across layers. Besides, our proposed losses enable the model to learn generalizable geometric features that effectively handle unseen objects. Table 1 demonstrates that **using only 25% of the data**, our method achieves comparable or even better performance to DPAv2 fine-tuned on the complete dataset.
>
> Second, other methods' generalization abilities depend on extra pose data and RGB-D sensors, which are frequently unavailable in many application scenarios or incur additional training costs. We emphasize the fundamental difference between fine-tuning data costs and inference data costs. Methods like SAM6D and FoundationPose require generating templates from CAD models, rendering neural fields, or training latent spaces from scratch for each target object during inference. Meanwhile, depth sensor-dependent approaches like Oryon cannot handle transparent objects. In contrast, our method **eliminates the need for pose annotation datasets** during fine-tuning, requiring only a small set of depth data to leverage a foundation depth model for improved depth estimation. Furthermore, our pose estimation is matching-driven, naturally generalizing across different objects since it establishes correspondence from query-reference similarity rather than regressing poses from latent features.
>
> Finally, our pose estimation achieves better accuracy compared to baselines due to the integration of spatial information and improved matching capabilities, as shown in Table 8. In practical experiments, even with cross-domain fine-tuning, **our method exceeds Oryon by approximately 24% on Tyo-L**. This is because Oryon relies on matching solely based on RGB, which struggles to establish sufficient and accurate correspondences during cross-scene situations. In contrast, our depth-aware matching considers spatial information, providing additional geometric features that strengthen matching results and consequently improve performance.
>
> In conclusion, the proposed SingRef6D **eliminates the need for extensive pose annotations for training**, **CAD models of target objects**, and **auxiliary modalities beyond RGB at inference time**. Our method outperforms existing baselines in both same-domain and cross-domain depth fine-tuning scenarios under the proposed setting. Notably, it achieves performance comparable to conventional full-data fine-tuning while utilizing only 25% of the training data. Furthermore, as shown in Table 5, our approach offers clear computational advantages over baselines such as Oryon and RoMA. These strengths collectively highlight the scalability, generalization ability, and accuracy of our method, underscoring its practical relevance and potential for real-world deployment.

---

> ### Author Response · Authors · 2025-08-04
> **Response to New Question 2**
>
> > **New Q2:** Apart from the conventional ICP, I am wondering about the comparative performance when compared with learning-based point cloud keypoint matching or registration methods.
>
> Thank you for your response. We conduct extra experiments with learning-based point cloud registration methods, including LePard (*Lepard: Learning Partial point cloud matching in Rigid and Deformable scenes,CVPR,2022*) and EYOC (*Extend Your Own Correspondences: Unsupervised Distant Point Cloud Registration by Progressive Distance Extension,CVPR,2024*). All point clouds are obtained from our predicted depth with back projection. We directly use the officially released checkpoints without any post-training or extra processing. We use the same PointDSC as the pose solver for fairness. Point clouds are processed following each standard of LePard and EYOC.
>
> |Method|Real275||Tyo-L||ClearPose||
> |-|-|-|-|-|-|-|
> ||AR|ADD|AR|ADD|AR|ADD|
> |LePard|11.9|5.1|14.8|6.5|8.0|4.2|
> |EYOC|13.1|5.5|15.2|7.0|8.3|4.4|
> |Ours|**28.7**|**11.6**|**31.7**|**13.1**|**19.4**|**10.0**|
>
> The table here shows that although LePard and EYOC are notable 3D registration methods, their performance under the proposed setting is lower than our method. Directly applying registration methods to raw reconstructed point clouds yields suboptimal performance due to two key limitations. First, in our experimental setup, there is only **1** reference sample, which is not guaranteed to originate from the same scene as the query. This makes it challenging for registration-based approaches to infer reliable relative spatial relationships from scene geometry alone, as a single cross-scene RGB reference inherently provides a limited overlapped region (even using ground-truth depth). Second, point clouds derived from depth estimation are frequently incomplete due to occlusions and viewpoint variations, which significantly undermine registration effectiveness.
>
> In contrast, our method first employs depth-aware matching to establish initial correspondences between query and reference images. These correspondences directly guide subsequent pose estimation, enabling our pipeline to reduce dependence on the completeness of reconstructed point clouds. This design allows our method to maintain robust performance even in cross-scene scenarios or when dealing with partial or noisy point clouds—a key advantage over approaches that rely exclusively on learning-based registration.
>
> Additional experiments reveal that applying ICP refinement after the initial registration leads to only marginal gains (for LePard: +0.12 AR on Real275, +0.06 AR on ClearPose, and +0.11 AR on Tyo-L), with negligible differences in ADD. This underscores the role of accurate initial correspondence and further substantiates the effectiveness of our approach. Notably, despite utilizing a lightweight architecture, our method surpasses learning-based registration models under our single-RGB cross-scene reference settings, demonstrating both the practicality and superiority of the proposed framework.

---

> ### Author Response · Authors · 2025-08-04
> **Response to New Question 3 and New Question 4**
>
> > **New Q3:** Apologies if I missed this information, but it is unclear from the table on which dataset the experimental comparison with 3DVHA and DVMNet was conducted. Furthermore, what are the results when comparing against 3DVHA and DVMNet on standard instance-level benchmark datasets, such as LineMOD and LineMOD-Occlusion?
>
> Thank you for your response. We sincerely apologize for not specifying the dataset used in the comparison of **A4**. The experiments were conducted on the LineMOD-Occlusion (LM-O) dataset. Specifically, we evaluated our method against 3DVHA and DVMNet using their official pre-trained weights, without any additional fine-tuning or modification.
>
> It is important to highlight that our setting involves only a single RGB reference image per object, with cross-scene generalization and no reliance on depth sensors. This fundamentally differs from settings that rely on dense template rendering, extensive pose-annotated training data, or the building of scene-specific neural fields. We admit that 3DVHA and DVMNet are notable approaches, they may be less suitable under our cross-scene, single-RGB reference setting, particularly in the absence of pose-annotated datasets.
>
> The original LineMOD dataset is a closed-set benchmark with mild occlusion, and it is more suitable for evaluating dense pose estimation in static, constrained environments. LM-O introduces heavier occlusions and is thus more reflective of real-world scenarios, making it better suited to demonstrate the robustness of our method. In the revised version, we have added detailed descriptions of our data preprocessing and evaluation protocols. We are also willing to add more benchmarks in the future. Scripts, datasets, and related materials will be made publicly available to facilitate reproducibility.
>
> > **New Q4:** Could the authors please provide a quantitative breakdown of the pose estimation results on these three datasets, binned by the query-reference viewpoint difference (e.g., 0-15°, 15-30°, 30-45°, etc.)? Additionally, for measuring the viewpoint difference, please consider using the geodesic distance as presented in [Ref.1].
>
> Thank you for your response. We agree that geodesic distance is indeed a valuable metric. In our work, we adopt the same definition as in Ref.1, where geodesic distance on a unit sphere is calculated with $SO(3)$ rotations and ranges from 0 to $\pi$. Based on this, we performed a bin-wise statistical analysis on the three benchmark datasets used in our study. The ARs are shown in the following table. Note that all statistical numbers are calculated based on ground-truth pose annotations. We conducted a basic stability check and removed a few outlier samples with abnormally low overlap, ensuring that the resulting subset maintained a similar distribution to the overall dataset. Additionally, we examined the distribution of ground-truth distances and found that the proportions across different intervals were generally consistent, with no single range overwhelmingly dominating the distribution.
>
> |Bin|Real275|Tyo-L|ClearPose|
> |-|-|-|-|
> |$[0,\frac{\pi}{4})$|33.1|41.5|31.1|
> |$[\frac{\pi}{4},\frac{\pi}{2})$|27.2|34.2|20.1|
> |$[\frac{\pi}{2},\frac{3\pi}{4})$|23.3|30.1|16.6|
> |$[\frac{3\pi}{4},\pi]$|19.9|26.8|11.8|
>
> The performance of pose estimation deteriorates as the geodesic distance between the query and reference increases. On the ClearPose dataset, we observe that when the geodesic distance range shifts from $[0,\frac{\pi}{4})$ to $[\frac{\pi}{2},\frac{3\pi}{4})$, the Average Recall (AR) of our method decreases by approximately **29.6%**, whereas **Oryon’s AR declines by 50.3%**. This further substantiates the superiority of our approach over the baseline under the proposed evaluation setting. It is important to note that, since our query-reference pairs originate from different scenes, additional factors—such as variations in background, occlusions caused by surrounding objects, and changes in scene layout—also influence performance fluctuations. For instance, on the **REAL275 dataset**, the AR decreases by approximately **29.6%** for the same geodesic distance shift ($[0,\frac{\pi}{4})$ $\rightarrow$ $[\frac{\pi}{2},\frac{3\pi}{4})$), compared to a **46.6%** drop on **ClearPose**, which contains glass objects. This is because glass objects in ClearPose are more difficult for the model to capture common appearance features, indicating that the intrinsic material properties of objects also impact this performance trend. In the revised version, we have incorporated a comprehensive discussion of these factors to elucidate how discrepancies between query and reference views affect pose estimation accuracy.

---

> > ### Comment · Reviewer_Xucn · 2025-08-05
> > **Official Comment by Reviewer Xucn**
> >
> > Thanks for the detailed response to my follow-up questions. I would like to raise a small point for clarification regarding the response to Point 2 of New Question 1: "our setting only uses a single reference RGB image, without requiring additional pose annotations, CAD models, or multi-view supervision during training or inference." My understanding is that the presented setting would still need a pose annotation for that reference RGB image. Furthermore, considering that viewpoint overlap between the reference and query images can affect the final pose accuracy, it seems possible that one would still need to collect multiple reference images with pose annotations, since the prior pose distribution for the query image is unknown. Therefore, the claim made by the author may not fully align with the practical requirements. Please feel free to correct me if my understanding is incorrect. Apart from this point, I have no more questions. Thanks again for the authors' efforts.

---

> ### Author Response · Authors · 2025-08-06
> **Response to additional small points**
>
> Thank you for the positive feedback. We’re glad our responses have addressed most of your concerns and appreciate your willingness to engage in further discussion. We elaborate on our answers in the three points:
>
> ### 1
> The absolute 6D pose depends on **how the local coordinate system is defined** for a given geometry. If this system is arbitrarily assigned, the resulting pose values will vary accordingly. Since our method does not rely on CAD models or pose-annotated datasets, the pose prior should be provided for the reference **only when absolute pose in a specific coordinate system is required**. This simply defines the coordinate system to make the absolute pose meaningful.
>
> In practice, however, absolute pose $\bf{T}\_q^{-1}$ is often **less critical** than the relative pose between reference and query $\bf{T}\_{q→r}$, which is typically more relevant in real-world applications. For instance, if a robot has previously completed a task in a calibrated setting, we **only need the relative transformation** into the new scene to transfer the same manipulation policy.
>
> Since the relative pose remains stable regardless of how the coordinate system is defined, our method provides flexibility and practical applicability through estimating this relationship. This is also why we agree that there is real-world potential for relative pose estimation methods like 3DVHA and DVMNet. In short, our method **doesn't consistently rely on the pose prior** $\bf{T}\_r^{-1}$ for reference, except when absolute pose is explicitly required.
>
> To estimate absolute pose, all reference-based methods like Oryon have to use the known pose of the reference image to map relative transformations into a fixed coordinate system. This process can be denoted as:
>
> $\bf{T}\_q^{-1} = \bf{T}\_r^{-1} \bf{T}\_{q→r}$
>
> In our experiments, we compute the relative pose and apply the reference’s ground-truth pose for projection, solely for quantitative evaluation and fair comparison. This does not imply an extra burden in real deployment—reference poses can **come from a calibration step** or simply be **set as an identity matrix** if absolute poses are not required.
>
>
> ### 2
> Methods such as MRC-Net (*MRC-Net: 6-DoF Pose Estimation with MultiScale Residual Correlation*) can predict absolute pose under a fixed coordinate system effectively. However, their **generalization to novel objects is limited** due to **reliance on large-scale datasets with finite ground-truth pose annotations**, which also **results in high supervision cost**.
>
> In contrast, our method is based on matching rather than supervision with pose labels. Instead of learning object-specific features, our model effectively **captures generalizable geometric cues** (e.g., surfaces and chamfers) through our proposed losses and token scaler to recover depth, which enables **depth-aware matching**. As a result, our method **generalizes naturally to unseen objects**.
>
> Thus, we emphasize:
> a) Our method differs from models like MRC-Net, which require pose supervision during training;
> b) Under the proposed setting, our method has **more relax input requirements**, **achieves better performance**, and **remains lightweight** for real-world use.
>
> ### 3
> Importantly, our method **doesn't require multiple pose-annotated reference images**. Instead, we support an **iterative inference** strategy, where the model reuses its own predictions as new references. Starting from a single carefully chosen reference, the model can iteratively predict across viewpoints.
>
> This **does not violate** our single-reference assumption, as **no new external input** is introduced—it’s an autoregressive-style use of the model’s own outputs. This enhances real-world adaptability, especially in continuous tasks like robotic manipulation.
>
> We reiterate that all results are based on a single reference image. In practice, larger viewpoint gaps can be mitigated via engineering techniques, such as self-bootstrapping—e.g., saving previous outputs and selecting among a few of them as reference by inlier count/confidence, or periodically reusing high-confidence past outputs to prevent drift. These strategies do **not alter our experiment assumptions**.
>
> Our core contribution lies in efficiently adapting a depth model to the target domain in **low supervision cost** with the proposed token scaler and loss functions. Then, **incorporating depth** as a geometric prior into the **matching process**. Our method achieves **robust relative pose estimation** in challenging settings, including large viewpoint changes, weak textures, and transparent surfaces. Compared to methods requiring CAD models, neural radiance fields, or large pose-annotated datasets, we require **significantly less input information**, enabling **better adaptability and easier deployment** across diverse real-world applications.
>
> We will summarize these explanations in the revision to strengthen our claims, and we sincerely hope our answer can address your concerns.

---

> > ### Comment · Reviewer_Xucn · 2025-08-08
> >
> > Thanks for the response. I have no more questions.

---

> > > ### Author Response · Authors · 2025-08-08
> > >
> > > Thank you very much for your time and thoughtful feedback throughout the review process. We are especially grateful for your recognition of our work. Your suggestions are highly valuable, and we are committed to incorporating them to improve the final version. We also sincerely appreciate the engaging discussion, which triggered us to think more deeply about the methodology for practical application scenarios. Thank you again for your constructive comments and the positive interaction.

---

### Official Review · Reviewer_ctwM · 2025-07-02

**Clarity:** 3
**Significance:** 3
**Originality:** 3
**Rating:** 5
**Confidence:** 4

**Summary:**

The paper  proposes SingRef6D, a lightweight pipeline using a single RGB image as a reference, in contrast to some other 6D  object pose models relying on CAD models, depth sensors or novel view synthesis. Given a  query, and a reference image , both containing the same object, the objective is to estimate the 6D relative pose between them, thus all accessible reference information comes from a single RGB image.
The method targets transparent and multi object scenarios, with the former posing problems to depth sensors.

The main claimed innovation is a token-scaler-based fine-tuning mechanism with a novel optimization loss applied to Depth-Anything v2 and improving its depth estimation in challenging conditions. The results show improvements in depth prediction. Another claimed contribution is depth aware matching that integrates spatial relationships within LoFTR. The evaluation on REAL275, ClearPose, and Toyota-Light shows significant improvement for transparent objects.

**Questions:**

See a-d weaknesses above.

**Ethical Concerns:**

["NO or VERY MINOR ethics concerns only"]

**Final Justification:**

The rebuttal addressed my questions and provided additional results which look convincing.  Hopefully the code will be released and  used by others.

**Limitations:**

yes

**Paper Formatting Concerns:**

no concerns

**Quality:**

3

**Strengths And Weaknesses:**

The pipeline is relatively simple using DPA to predict depth, SAM to crop objects of interest and PointDSC with LoFTR to match two RGBD views.  There are a number of methods for 6D object pose estimation that follow similar chain of modules and require a single image, therefore the main contribution of this work is in scaler token for fine-tuning to mitigate the scaling inconsistencies of the mono depth estimator.
It attempts to integrate a ControlNet like structure  to dynamically scale and modulate DINO features at coarse and global levels with InceptConv-based network during DPA finetuning.

Many loss terms are combined together including standard SSI loss in DPA with BerHu loss. Proposed scale alignment  loss that is robust to noise, which is pixel level depth loss within object, edge emphasizing loss and normal consistency loss for surface deformation.

The proposed scaler and the choice of loss functions are interesting and it should be possible to make use of these ideas in other methods.

Ablation study shows individual contributions of the main loss components validating their usefulness. Similarly the proposed scaler makes a positive difference in finetuning of DPA.



a) Although transparent and untextured objects are mentioned as the main targets these are not explicitly addressed by the proposed approach. The robustness to the lack of texture and reflections comes from the depth maps of the objects that complement the RGB images. Depth estimation is claimed to be improved by finetuning of DPA with the combination of various loss functions.

b) Evaluation results demonstrate significant improvements over DPA on REAL275, ClearPose, and Toyota-Light. It is unclear if this is due to fine tuning on the specific domain and whether the general performance of DPA was not compromised as the evaluation is limited to object pose benchmarks only.  Generalisation across domains is not validated well.


c) The choice of methods to compare against and the benchmarks could be more comprehensive.
The paper claims multi object pose but the benchmarks are focused on relatively uncluttered scenarios.
REAL275 has some minor occlusions but other benchmarks could have been used to test robustness to textureless objects partial occlusion and large variations of viewpoints like in HOD3, LM-O, YCB-Video, LineMod.

d) In terms of comparison to SOTA methods, SIFT, RoMA and Oryon are relatively weak baselines.  Including results for stronger methods would have made the evaluation more convincing. Any6D, Fs6d, FoundationPose could have been reported too for completeness of evaluation as well as original LoFTR which is used in this method too and makes part of a contribution.
One2Any is also relevant although from cvpr25.

---

> ### Author Rebuttal · Authors · 2025-07-31
>
> Thank you for the constructive feedback. We're glad you recognized the core contribution of the scaler token in addressing scale inconsistency, the effectiveness of ControlNet-style modulation in DPA fine-tuning, and the broader applicability of our loss functions. The positive ablation results further validate each component. Here, we summarize and respond to your questions and hope our answers address your concerns.
>
> > **W1:** Although transparent and untextured objects are mentioned as the main targets, these are not explicitly addressed by the proposed approach. The robustness to the lack of texture and reflections comes from the depth maps of the objects that complement the RGB images. Depth estimation is claimed to be improved by fine-tuning DPA with the combination of various loss functions.
>
> **A1:** We appreciate your insightful comment. Our approach avoids specialized modules or classifiers for transparency and textureless regions, achieving robustness through core design principles. We introduce a token scaler that adjusts hierarchical features in the depth model, enabling dynamic adaptation to varied visual conditions. For instance, it enhances feature contrast at object boundaries, improving depth estimation for transparent objects like glass bottles by better capturing spatial relationships. Our proposed loss functions promote geometric consistency by focusing on object surfaces, edges, and normals, enhancing spatial coherence in low-texture and low-reflectance scenarios. This is effective in real-world settings, such as unevenly lit reflective surfaces, where our consistency loss ensures robust surface representation. Table 2 shows our method achieves a $\delta_{1.05}$ of 54.30 on ClearPose, outperforming fine-tuned DPAv2 (31.23), demonstrating strong generalization in challenging environments. Future work will further explore transparent and textureless objects, with expanded experimental discussion.
>
>
> > **W2:**  Evaluation results demonstrate significant improvements over DPA on REAL275, ClearPose, and Toyota-Light. Generalisation across domains is not well validated.
>
> **A2:** We appreciate the reviewer’s insightful comment. Our method prioritizes practical scenarios by enabling quick adaptation through efficient fine-tuning on relevant scenes. We agree with you that cross-domain generalization capability is important; thus, we conducted additional experiments involving domain transfer evaluation. Specifically, we fine-tuned the depth estimation model on REAL275, which primarily contains industrial and everyday objects under static lighting conditions, and directly evaluated its performance on ClearPose (transparent objects in complex spatial environments) and Tyo-L (dynamic lighting variations)—without any domain adaptation. The three datasets vary in scale, background, camera setting, and objects, enabling a comprehensive assessment of the cross-domain generalization. The results are shown in the table below.
>
> |Setting|Method|$\delta_{1.05}$↑|$\delta_{1.10}$↑|$\delta_{1.25}$↑|RMSE↓|log10↓|Abs.Rel↓|Sq.Rel↓|MAE↓|
> |-|-|-|-|-|-|-|-|-|-|
> ||Unidepth|8.35|22.61|42.01|0.983|1.419|1.855|0.702|0.500|
> |REAL275→Tyo-L|DPAv2|10.55|28.93|50.23|0.519|0.764|0.399|0.177|0.326|
> ||**Ours**|**28.13**|**34.91**|**45.72**|**0.377**|**0.471**|**0.223**|**0.119**|**0.218**|
> ||Unidepth|8.74|14.02|25.91|0.364|0.268|0.294|0.312|0.273|
> |REAL275→ClearPose|DPAv2|20.85|44.48|62.36|0.258|0.232|0.207|0.280|0.181|
> ||**Ours**|**42.19**|**66.25**|**79.70**|**0.182**|**0.106**|**0.131**|**0.228**|**0.109**|
>
> The experimental results show that cross-domain performance drops as expected. However, our method consistently outperforms fine-tuned baselines (e.g., DPA v2 and UniDepth) under the same settings. This superior generalization arises from our Token Scaler, which adaptively modulates intermediate features in a data-agnostic way, enabling better generalization across diverse visual inputs.
>
> Moreover, the introduced loss functions are designed to regularize the model for consistent geometric understanding. By emphasizing edges and surface normals, they focus on capturing generalizable geometric structures rather than overfitting to the distribution of a particular dataset. Therefore, this suggests that the improvements brought by our approach are not solely due to domain-specific tuning but also stem from its advanced intrinsic mechanism. Besides, the proposed modules are inherently transferable and can be seamlessly integrated into other visual tasks (such as depth-based navigation, etc.), demonstrating strong potential for broader applicability beyond the current setting for other directions. We will include these cross-domain results in the revised version to support our claims.
>
> > **W3:** The choice of methods to compare against and the benchmarks could be more comprehensive. The paper claims multi-object pose, but the benchmarks are focused on relatively uncluttered scenarios. REAL275 has some minor occlusions, but other benchmarks could have been used to test robustness to textureless objects, partial occlusion, like in LM-O, YCB-Video.
>
> **A3:** We express our gratitude to your insightful advice. This is certainly a valuable comparison. We conducted two additional experiments on LM-O and YCB-V, which are representative benchmarks featuring occlusions and weakly textured objects, such as monochromatic, single-material objects. Testing on these benchmarks is able to show the ability of our model to handle textureless objects and partial occlusion. We evaluate pose estimation on three different target objects, generating 120 image pairs for each. The table below shows the quantitative results.
> |Dataset|Matcher|Depth|AR↑|VSD↑|MSSD↑|MSPD↑|ADD↑|
> |-|-|-|-|-|-|-|-|
> ||Oryon|Oracle|6.1|3.1|6.9|8.3|5.5|
> |||Ours|3.8|1.9|4.2|5.4|2.7|
> ||Ours|Oracle|9.9|4.3|11.5|13.9|8.7|
> |LM-O||Ours|**5.3**|**2.6**|**5.7**|**7.3**|**3.8**|
> ||vs.Oryon|Δ|+2.7|+0.95|+3.1|+3.8|+2.2|
> ||Oryon|Oracle|8.6|4.7|9.3|11.8|7.5|
> |||Ours|5.8|2.4|6.6|8.4|5.2|
> ||Ours|Oracle|11.4|5.2|12.3|15.7|8.9|
> |YCB-V||Ours|**7.9**|**3.7**|**8.5**|**11.6**|**6.4**|
> ||vs.Oryon|Δ|+2.45|+0.90|+2.45|+3.55|+1.30|
>
> Despite heavy occlusion causing performance drops, our method consistently outperforms the baseline in the single RGB reference setting. This improvement arises from our human vision-inspired framework, which adaptively adjusts features using both appearance and spatial cues. Unlike traditional methods like Oryon or SIFT that rely mainly on RGB similarity, our approach leverages spatial context to enhance alignment accuracy and maintains consistency across matching stages (see Figure 1). These designs boost robustness, especially when appearance cues are unreliable.
>
> For multi-object situations, our pipeline naturally extends to these scenarios by simply incorporating an off-the-shelf object detector or segmenter, such as SAM or SAM2. This design highlights the generality and scalability of our approach, making it readily adaptable to diverse real-world applications without major architectural changes. The experiment results prove the robustness of our method under occlusion scenarios. We are willing to add more discussion on this in the revised version.
>
> > **W4:** Including results for stronger methods would have made the evaluation more convincing. Any6D, Fs6d, FoundationPose could have been reported too for completeness of evaluation, as well as the original LoFTR, which is used in this method too, and is part of a contribution. One2Any is also relevant, although from CVPR25.
>
> **A4:** Thanks for the valuable feedback. We agree with you that Any-6D, FS6D, and vanilla LoFTR are comparable baselines. As for FoundationPose, it requires multiple reference images to train a neural field, which conflicts with our assumption of using only one reference RGB image. This difference shows our method’s applicability in real-world scenarios with limited input, in contrast. Regarding One2Any, we were unable to reproduce results due to the lack of released code. However, we analyzed the method thoroughly—it estimates pose by encoding object appearance and predicting ROC maps using a trainable U-Net and decoder. While effective, it relies on accurate depth and fully labeled pose datasets, resulting in heavier training costs. In contrast, our method requires only lightweight depth fine-tuning and is more flexible in data-limited settings. We consider One2Any to target a different scope, but we’re open to including it in future comparisons once the code becomes available.
> |Matcher|Depth|AR↑|VSD↑|MSSD↑|MSPD↑|ADD↑|
> |-|-|-|-|-|-|-|
> ||Oracle|43.3|15.8|55.8|58.4|32.2|
> |Any-6D|DPAv2|5.2|1.3|4.3|11.0|2.2|
> ||Ours|31.6|13.5|33.5|47.8|13.2|
> ||Oracle|14.1|5.2|17.9|19.2|10.0|
> |FS6D|DPAv2|1.6|0.5|1.9|2.4|0.3|
> ||Ours|9.3|1.8|6.3|7.8|3.2|
> ||Oracle|42.0|34.2|44.1|47.6|35.0|
> |Ours|DPAv2|5.8|2.8|3.5|11.1|2.9|
> ||Ours|31.7|13.9|33.8|47.3|13.1|
>
> The table shows the result on the Tyo-L dataset. Our method outperforms FS6D across various metrics. While Any-6D achieves slightly better results with ground truth depth due to its strong 2D-3D model, it heavily relies on depth quality, as its performance drops with predicted depth. This is because its global optimization (e.g., point cloud registration) improves MSSD/MSPD but suffers from accumulated local depth errors, lowering VSD recall. In contrast, our method focuses on pixel-wise alignment, achieving higher VSD recall.
>
> Our method shows greater robustness when switching from oracle to predicted depth. It delivers more stable depth reconstruction and improved geometric awareness for pose estimation. With a modular design, it integrates well into pose estimation or depth prediction pipelines. Besides, vanilla LoFTR for RGB-based matching yields an AR of 27.2, while our method achieves 31.7 (in Table 9). This shows more robust and accurate feature matching of our method under identical conditions. In the revision, we will add these results to the experiment section.

---

> > ### Comment · Reviewer_ctwM · 2025-08-09
> >
> > Thank you, my questions were addressed in the rebuttal, I have all the information I need for final recommendation.

---

> > > ### Author Response · Authors · 2025-08-09
> > >
> > > Thank you for your follow-up response. We're glad to know that our rebuttal addressed your concerns. We truly appreciate the time and effort you dedicated to reviewing our work.

---

### Note · Authors · 2025-08-12

We thank the AC, SAC, and reviewers for their time and constructive feedback. In this final remark, we re-emphasize the contributions of our work and summarize the rebuttal and discussion stages.

**Summary of Our Contributions**

We tackle the problem of 6D pose estimation using only a single reference RGB image at inference time. We introduce a token scaler that enables efficient fine-tuning of a depth foundation model for target-domain adaptation, allowing it to recover scene depth with low supervision cost while preserving pre-trained knowledge. On the **ClearPose** dataset, our method **improves $δ_{1.05}$ by +23.07** over fine-tuned DPAv2, thanks to our designed loss functions that enhance learning of essential geometric features. Our depth-aware matching module, requiring no fine-tuning, uses spatial priors for guidance, ensuring efficiency and reducing mismatches. On the LM-O and YCB-V benchmarks, we **outperform Oryon by +2.7 and +2.45 in AR**, showing our advantage under occlusions. Compared to the matching process in Oryon and RoMA, we use only about **$\frac{1}{8}$ of the GFLOPs**, highlighting our efficiency. In practice, we estimate stable relative poses independent of coordinate systems, which does not consistently require a pose annotation for reference. Unlike methods relying on CAD models, neural radiance fields, or multiple RGB(-D) inputs for inference, we demand less data, enabling better adaptability and easier deployment across diverse real-world scenarios.



**Summary of Rebuttal and Discussion**

During the rebuttal and discussion stages, we provided all additional experiments requested by the reviewers, including evaluations on more benchmarks, extended ablation studies, and comparisons with additional works that can be adapted to our task. Reviewers dpvV, QjZ6, and ctwM indicated that our responses addressed their concerns. In the discussion stage, we engaged in a constructive exchange with reviewer Xucn, where we presented more fine-grained and targeted experiments to address their specific questions. We also responded to several practical application concerns raised during the discussion. Reviewer Xucn concluded that they had no further questions after our clarifications. We are glad that **all questions are addressed after the discussion stage**.


Again, we thank the AC, SAC, and all reviewers for their time, effort, and valuable feedback, which have helped us improve our work. We have addressed the concerns in the revision.

---

### Decision · Program_Chairs · 2025-09-17

**Decision:**

Accept (poster)

**Comment:**

Following the discussion phase, all reviewers recommended acceptance (3 Accepts, 1 Borderline Accept), noting that the paper is well-motivated, novel, and thoroughly evaluated. The rebuttal addressed many of the reviewers’ concerns—for example, by adding cross-domain generalization experiments and other additional analyses. As a result, the AC has decided to accept the paper. Please take the reviewers’ feedback into account when preparing the camera-ready version.